# Structural basis for diguanylate cyclase activation by its binding partner in *Pseudomonas aeruginosa*

Gukui Chen[1†], Jiashen Zhou[2†], Yili Zuo[1], Weiping Huo[1], Juan Peng[1], Meng Li[1], Yani Zhang[1], Tietao Wang[1], Lin Zhang[2], Liang Zhang[2]*, Haihua Liang[1,3]*

[1]Key Laboratory of Resources Biology and Biotechnology in Western China, Ministry of Education, College of Life Sciences, Northwest University, ShaanXi, China; [2]Department of Pharmacology and Chemical Biology, Shanghai Jiao Tong University, School of Medicine, Shanghai, China; [3]School of Medicine, Southern University of Science and Technology, Shenzhen, China

**Abstract** Cyclic-di-guanosine monophosphate (c-di-GMP) is an important effector associated with acute-chronic infection transition in *Pseudomonas aeruginosa*. Previously, we reported a signaling network SiaABCD, which regulates biofilm formation by modulating c-di-GMP level. However, the mechanism for SiaD activation by SiaC remains elusive. Here we determine the crystal structure of SiaC-SiaD-GpCpp complex and revealed a unique mirror symmetric conformation: two SiaD form a dimer with long stalk domains, while four SiaC bind to the conserved motifs on the stalks of SiaD and stabilize the conformation for further enzymatic catalysis. Furthermore, SiaD alone exhibits an inactive pentamer conformation in solution, demonstrating that SiaC activates SiaD through a dynamic mechanism of promoting the formation of active SiaD dimers. Mutagenesis assay confirmed that the stalks of SiaD are necessary for its activation. Together, we reveal a novel mechanism for DGC activation, which clarifies the regulatory networks of c-di-GMP signaling.

*For correspondence:
liangzhang2014@sjtu.edu.cn
(LZ);
lianghh@nwu.edu.cn (HL)

†These authors contributed equally to this work

Competing interest: The authors declare that no competing interests exist.

## Introduction

The intracellular messenger cyclic dimeric (3'–5') guanosine monophosphate (cyclic di-GMP or c-di-GMP) is a nearly ubiquitous bacterial second messenger that mediates various physiological aspects of diverse environmental and pathogenic bacteria (*Jenal et al., 2017*; *Römling et al., 2013*). It was first described in 1986 as an allosteric factor that activated cellulose synthase in *Acetobacter xylinum* (*Ross et al., 1987*). To date, c-di-GMP has been shown to regulate the cell cycle, biofilm formation, dispersion, motility, virulence, and other processes (*Jenal et al., 2017*; *Römling et al., 2013*) as well as to promote community interactions during biofilm growth in many Gram-negative bacteria by stimulating the production of exopolysaccharide and adhesins (*Jenal et al., 2017*; *Simm et al., 2004*). The intracellular c-di-GMP level is controlled by a dynamic balance of synthesis by diguanylate cyclases (DGCs, containing a GGDEF domain) and degradation by specific phosphodiesterases (PDEs, containing EAL or HD-GYP domains) (*Kulasakara et al., 2006*; *Schmidt et al., 2005*; *Simm et al., 2004*). Degradation of c-di-GMP by PDEs containing an HD-GYP or EAL domain results in the production of GMP or 5-phosphoguanylyl-(3',5')-guanosine (pGpG), which is further hydrolyzed by oligoribonucleases into two GMP molecules (*Cohen et al., 2015*; *Orr et al., 2015*).

Multiple DGCs with characteristic GG(D/E)EF domains produce c-di-GMP in response to a variety of environmental stimuli due to the presence of specialized sensory or regulatory domains (*Schirmer, 2016*). Diverse molecular mechanisms have been reported for the activation of DGCs in several bacterial species. Phosphorylation of a DGC by a related kinase promotes its activity, as reported for PleD

in *Caulobacter crescentus* (*Paul et al., 2007*) and WspR in *Pseudomonas aeruginosa* (*Hickman et al., 2005Hickman et al., 2005*). The activity of some DGCs is regulated by a partner-switch system (PSS). For example, the activities of both BgrR in *Sinorhizobium meliloti* (*Baena et al., 2017*) and HsbD in *P. aeruginosa* (*Valentini et al., 2016*) are regulated by a STAS domain protein (BgrV or HsbA). Notably, the phosphorylation status of the STAS domain protein is essential to its control of DGC activity. Recently, we elucidated that the DGC activity of SiaD is modulated by its binding partner via direct interaction, whereas phosphorylation of SiaC by the kinase SiaB prevents the SiaC-SiaD interaction and activation of SiaD (*Chen et al., 2020*). Other mechanisms for DGC activation have been also reported. In *Escherichia coli*, the DGC DgcZ is regulated allosterically by zinc (*Zähringer et al., 2013*). In addition, DgcZ is a substrate for the Sir2 family protein deacetylase CobB. Deacetylation of DgcZ by CobB enhances its activity and thus promotes c-di-GMP production (*Xu et al., 2019*). Additionally, the presence of oxygen regulates the DGC activity of SadC via the proteins OdaA and OdaI (*Schmidt et al., 2016*).

In *P. aeruginosa*, the SiaA/B/C/D signaling network regulates biofilm and aggregate formation in response to environmental stimuli. Although we revealed that the DUF1987 domain-containing protein SiaC significantly promotes the DGC activity of SiaD (*Chen et al., 2020*), the underlying mechanism for SiaD activation remains unclear. In this study, we provide structural, biochemical, and in vivo data to elucidate how the activity of SiaD promoted by its binding partner SiaC. The widespread distribution of DUF1987 domain proteins indicates the significance of this activation mechanism. Hence, we reported a novel and atypical mechanism for DGC activation by its binding partner.

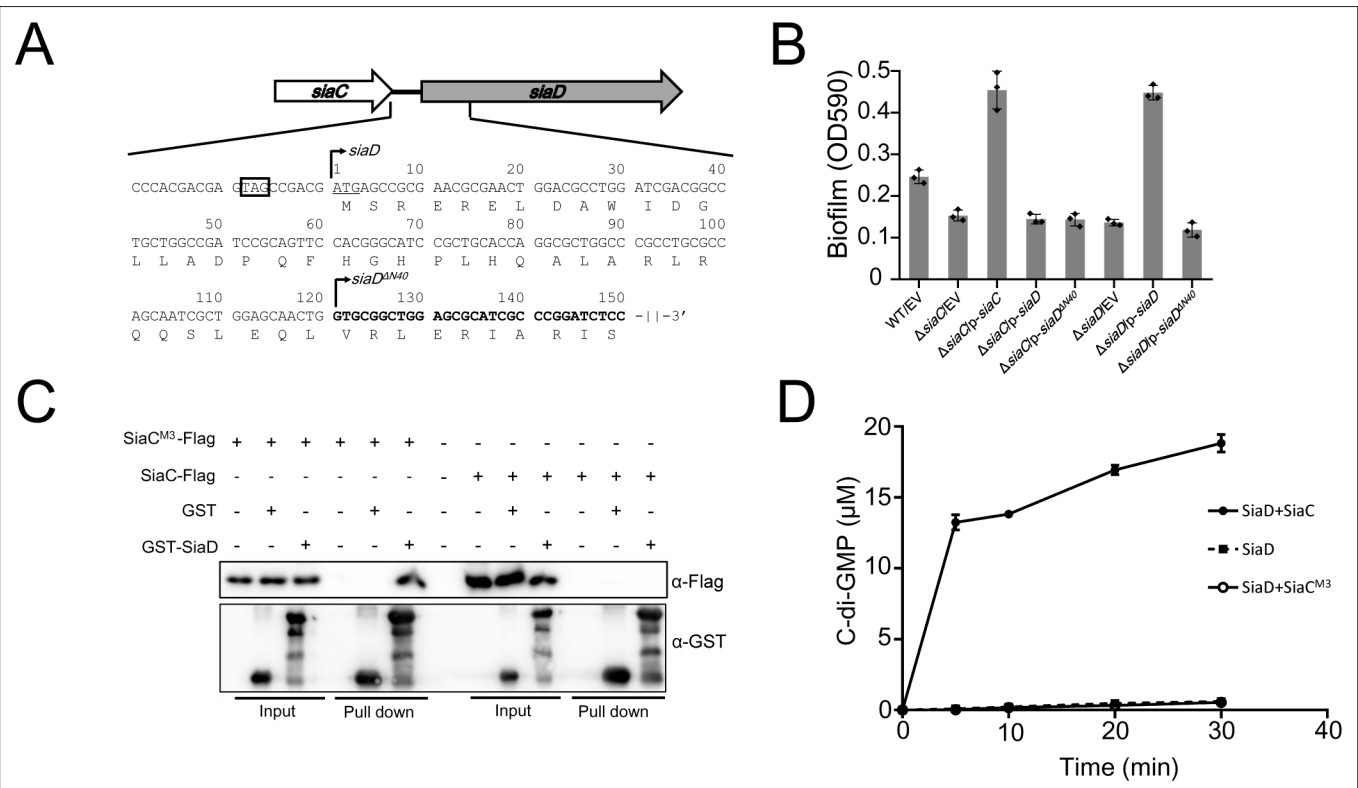

**Figure 1.** SiaC promotes the DGC activity of SiaD through direct interaction. (**A**) Analysis of the initiation codon and coding sequence of SiaD. (**B**) Overexpression of SiaD but not SiaD^ΔN40 restored the biofilm formation of *siaD* mutant. The biofilm formation of the indicated strains was stained by crystal violet staining and quantified with optical density measurement. (**C**) Pull down assay revealed the role of N67/T68/S69 for SiaC to interact with SiaD. Cell lysates of *E. coli* containing pMMB67EH-*siaC*-Flag or *siaC*^N67A/T68A/S69A-Flag (*siaC*^M3-Flag) were incubated with GST or GST-SiaD, individually, and protein complexes were captured by glutathione beads. (**D**) Production of c-di-GMP by SiaD, SiaD with SiaC, and SiaD with SiaC^N67A/T68A/S69A (SiaC^M3).

The online version of this article includes the following figure supplement(s) for figure 1:

**Figure supplement 1.** SiaD coding sequence.

**Figure supplement 2.** SDS–PAGE for purifed SiaD and SiaD^ΔN40 proteins.

**Figure supplement 3.** Production of c-di-GMP by SiaD, SiaD^ΔN40 determined by HPLC.

## Results

### The coding sequence of SiaD is redefined and its activity is promoted by the binding partner SiaC via direct interaction

*siaD* is co-transcribed with *siaABC* (*Klebensberger et al., 2009*). The *siaD* gene has been annotated as a 708 bp-length gene initiated with a GTG codon (https://www.pseudomonas.com/) (*Klebensberger et al., 2009*; *Winsor et al., 2016*). However, we analyzed the 125 bp intergenic region between *siaC* and *siaD* and revealed an ATG initiation codon. Therefore, we speculated that the coding sequence of *siaD* begins at this ATG codon upstream of the previously annotated GTG codon. To confirm the coding sequence of the *siaD* gene, two plasmids, overexpressing *siaD* or *siaD*$^{\Delta N40}$, were constructed (pMM-*siaD* and pMM-*siaD*$^{\Delta N40}$) (*Figure 1A*). A biofilm assay showed that the plasmid overexpressing *siaD*$^{\Delta N40}$ failed to restore biofilm formation in the Δ*siaD* mutant. However, biofilm formation in the Δ*siaD* mutant was fully restored by SiaD overexpression (*Figure 1B*). This result suggests that the longer SiaD sequence, but not SiaD$^{\Delta N40}$, is the functional form of this protein in vivo. Therefore, we refer to SiaD as the longer protein at the ATG initiation codon henceforth.

Previously, we demonstrated that the function of SiaD was dependent on SiaC in vivo and revealed that SiaC acts as a binding partner regulating the DGC activity of SiaD$^{\Delta N40}$ in vitro via direct protein–protein interaction. To determine the relationship between SiaC and SiaD, a glutathione-S-transferase (GST) pull-down assay was performed using purified GST-SiaD. The results showed that SiaC was retained with GST-SiaD, but not with GST (*Figure 1C*), confirming the SiaC–SiaD interaction. This observation led us to explore whether SiaC influences the stability of SiaD in vivo. To this end, wild-type PAO1 and Δ*siaC* mutant with the pMMB67EH-*siaD*-flag plasmid were cultured in the presence of 1.0 mM IPTG for 1 hr; 50 μg/ml spectinomycin was added to the medium to block protein synthesis. Samples of bacterial culture were collected at 0, 0.5, 1.0, 1.5, 2.0, and 3.0 hr and subjected to western blotting. Data showed that the stability of SiaD was not influenced by deletion of *siaC* (*Figure 1—figure supplement 1*).

Furthermore, the DGC activity of SiaD (*Figure 1—figure supplement 2*) was evaluated in vitro by measuring the production of c-di-GMP from guanosine triphosphate (GTP) using high-performance liquid chromatography (HPLC) analysis, which showed retention times consistent with those of the GTP and c-di-GMP standards. DGC activities of SiaD both in the presence and absence of SiaC were determined. Consistent with the results for SiaD$^{\Delta N40}$, little c-di-GMP was produced when GTP was incubated with SiaD alone. In the presence of SiaC, the production of c-di-GMP increased sharply, suggesting that SiaC promotes the DGC activity of SiaD via direct interaction (*Figure 1D*). Interestingly, we found that in the presence of SiaC, the DGC activity of SiaD was significantly greater than that of SiaD$^{\Delta N40}$ (*Figure 1—figure supplement 3*), which further highlights the importance of amino acids 1–40 of SiaD and accounts for the inability of *siaD*$^{\Delta N40}$ to promote biofilm formation. As SiaC was unable to produce c-di-GMP or enhance the activity of another DGC, WspR, we concluded that interaction with SiaC is essential for the DGC activity of SiaD.

### Overall structure of the SiaC–SiaD complex reveals a unique binding pattern

To investigate the regulation of SiaD by SiaC, we determined the crystal structure of full-length SiaD in complex with SiaC. The diffraction data was integrated and scaled using the program HKL3000 to space group C222$_1$ at 2.65 Å resolution, and the structure was subsequently determined through molecular replacement using the published conserved DGC domain of the WspR structure (pdb code: 3BRE) and the SiaC structure that we published previously (PDB code: 6KKP) (*Supplementary file 1c*). In the asymmetric crystallographic unit, two SiaD and four SiaC molecules bind together, adopting a unique extensive conformation: two SiaD molecules (SiaD-A and SiaD-B) form a parallel coiled-coil via their stalk helices, and their dimeric stalks are stabilized by the binding of two pairs of SiaC molecules (SiaC-C/D, SiaC-E/F), each at distinct locations (proximal or distal to the SiaD DGC domain) along the dimeric stalk (*Figure 2A*). Moreover, a non-hydrolyzable GTP analog GpCpp molecule was observed to bind in the active site of SiaD-A.

In this complex, each the SiaD monomer consists of an N-terminal domain, a C-terminal DGC domain, and a long extended coiled-coil-like helical stalk domain linking the N- and C-terminal domains. The stalk and DGC domains of SiaD monomers bind each other in parallel, with a central

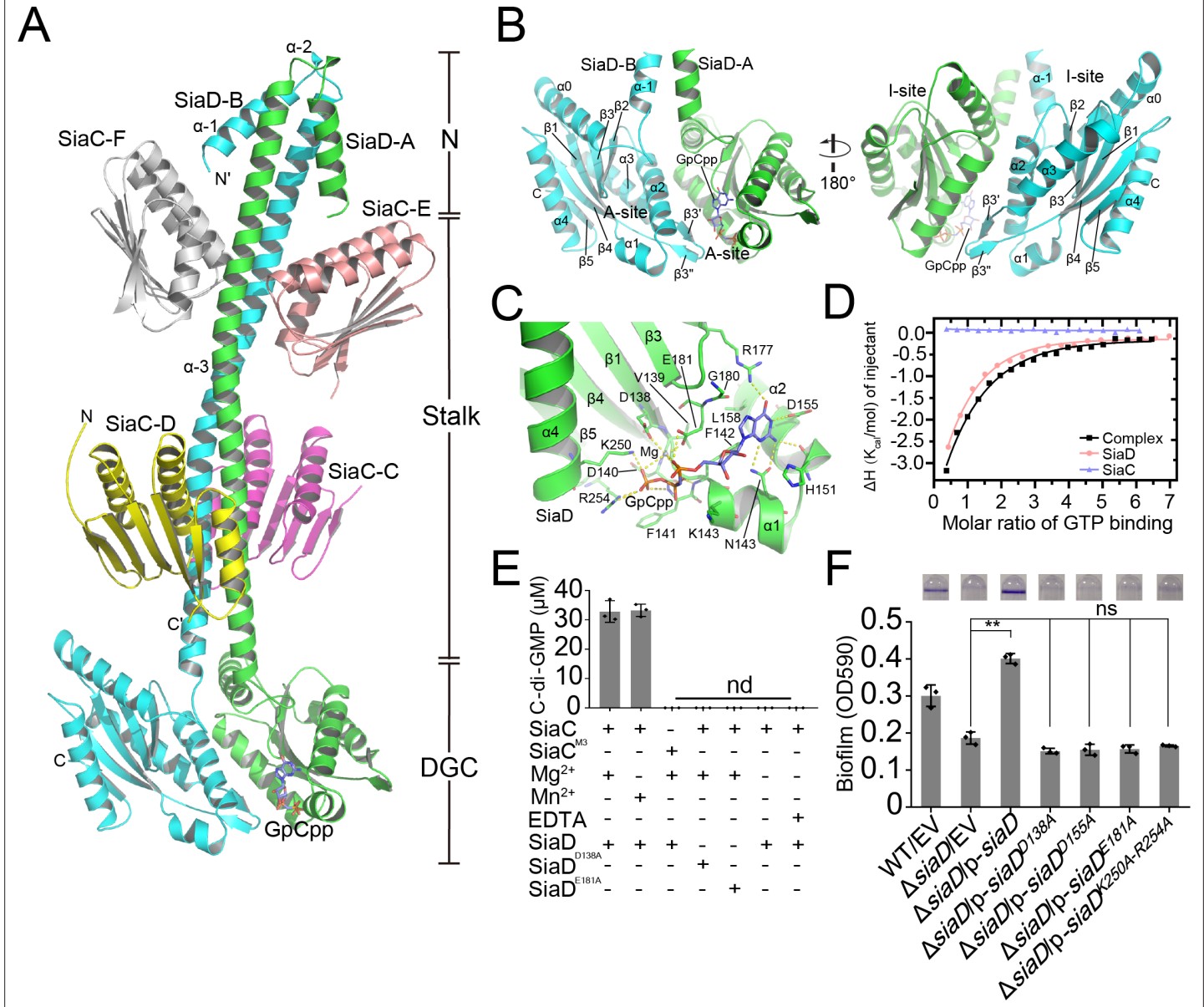

**Figure 2.** Crystal structure of SiaC–SiaD complex with GTP analog. Two monomers of SiaD (SiaD-A and SiaD-B) and four SiaC molecules (SiaC-C/D, SiaC-E/F) are colored in green, cyan, yellow, magenta, salmon, and white, respectively. The secondary structural elements of SiaD was labeled. The GTP analog GpCpp and $Mg^{2+}$ ion are shown in slate stick and lightblue sphere. H-bond network is shown as yellow dashes. (**A**) Overall crystal structure of SiaC–SiaD complex. The N-terminal domain, middle stalk domain, and C-terminal DGC domain of SiaD are labeled. (**B**) Overview of the DGC domain dimer. A-site and I-site were labeled, respectively. (**C**) GpCpp binds to the C-terminal DGC domain of SiaD (A-site). (**D**) Results of isothermal titration calorimetry (ITC) for GTP binding to SiaD (salmon), SiaC (slate), or SiaC–SiaD complex (black). Plots of molar enthalpy change against GTP-SiaD or SiaC–SiaD complex molar ratio are shown. (**E**) Divalent ion $Mg^{2+}$ or $Mn^{2+}$ is essential for SiaD activity. (**F**) Amino acid residues involve in GTP and $Mg^{2+}$ binding are essential for SiaD activity in vivo. The biofilm formation of the indicated strains were displayed with crystal violet staining (up) and quantified with optical density measurement (down). Data represent the means and SDs of three biological replicates. **$p<0.01$ based on one-way ANOVA test; ns, non-significance.

The online version of this article includes the following figure supplement(s) for figure 2:

**Figure supplement 1.** Multiple structural alignment of SiaD homologs.

**Figure supplement 2.** Superposition of SiaD with its homolog.

**Figure supplement 3.** Analysis of SiaD A-site with GpCpp.

**Figure supplement 4.** The stereo view of omit (fofc) electron density of GpCpp and $Mg^{2+}$ bound to SiaD at 3.0 σ.

**Figure supplement 5.** Expression of SiaD and its mutants during biofilm formation.

C2 axis of symmetry, to form a dimer that covers one-third of the SiaD protein surface (4460 Å$^2$ of 15,159 Å$^2$). In contrast to several DGCs (such as WspR and PleD), which employ a functional N-terminal CheY-like domain to specifically receive and respond to signals from upstream pathways, the N-terminal domain of SiaD has no such sensory domain. Instead, the two N-terminal helices (α–1 and α–2) are arranged in an antiparallel fashion with the dimeric coiled-coil composed of α–3 (*Figure 2A*). Although these short α-helices do not structurally interfere with the binding of SiaC, their deletion significantly diminishes the enzymatic activity of SiaD, suggesting that the N-terminus of SiaD plays a unique and critical role in dynamic catalytic and regulatory processes, rather than sensing upstream signals as reported in other DGCs.

## The DGC domain of SiaD is critical for substrate binding and allosteric inhibition

In contrast to the N-terminal domain of SiaD, the structure of the C-terminal GGDEF domain of SiaD is highly conserved throughout the DGC family (*Figure 2—figure supplements 1 and 2*; *Chan et al., 2004*; *De et al., 2008*). Two DGC domains in the complex are oriented anti-parallel, with their active sites facing each other (*Figure 2B*). Each DGC domain consists of a core canonical fold of five antiparallel β-strands (β1–β5) around which five α-helices (α0–α4) and two short antiparallel β-strands are wrapped (β3′ and β3"), forming a highly conserved GTP substrate-binding site (GGDEF domain, A-site) and a c-di-GMP product binding/inhibitory site (I-site).

The A-sites are located at one end of the GGDEF domain, and predominantly consist of residues from one SiaD monomer, hence there are two A-sites observed in the SiaD dimer complex. Structural and sequence alignment showed that residues involved in binding GTP and ion are highly conserved (*Figure 2—figure supplement 1*, *Figure 2—figure supplement 2* and *Figure 2—figure supplement 3A* ). However, there was only one GpCpp molecule observed in one of the SiaD A-sites, leaving the other A-site empty. Glutamate was the fourth amino acid of GGDEF, which was considered to be the 'general base' and can activate the nucleophile donor of another GGDEF monomer (*Hallberg et al., 2019*). By overlapping SiaD-A (containing GpCpp) with the GGDEF domains of *Is*PadC (a DGC structure with catalytic conformation, pdb code: 5LLX), it was observed that the conserved 'general base' E182 was still oriented toward the 3′ hydroxyl group from the GpCpp of the opposite monomer, which suggested cooperativity and conservation of SiaD as a catalytic dimer (*Figure 2—figure supplement 3B*). Inside the GpCpp bound A-site, GpCpp and an Mg$^{2+}$ ion are held in the catalytically competent position through multiple hydrophobic and hydrophilic interactions with residues inside the pocket (*Figure 2C*, *Figure 2—figure supplements 1 and 2* and *Figure 2—figure supplement 4*). Moreover, comparing structures of SiaD-A (with GpCpp) and SiaD-B, the interaction between Mg$^{2+}$/GpCpp and several key amino acids leads to deflection of the conformation (D138, E181, K250, and R254), which further stabilizes binding of Mg$^{2+}$ and the phosphate group of GpCpp (*Figure 2C* and *Figure 2—figure supplement 3C*). Multiple hydrogen and hydrophobic interactions between SiaD and GTP hold the ligand in the active pocket with a K$_d$ value of 34 μM, while SiaC alone does not bind GTP (*Figure 2D*). The divalent cation Mg$^{2+}$ is essential for DCG catalysis (*Chan et al., 2004*; *De et al., 2008*; *Jenal et al., 2017*). Consistently, we revealed that a divalent cation, Mg$^{2+}$ or Mn$^{2+}$, is necessary for SiaD activity (*Figure 2E*). Furthermore, a biofilm assay showed that mutation of key residues involved in binding GTP and Mg2$^+$ (D138A, D155A, E181A, and K250A/D254A) leads to loss of SiaD function (*Figure 2F*). Notably, these mutated forms of SiaD were well expressed during biofilm formation (*Figure 2—figure supplement 2*). We attempted to purify these SiaD mutant proteins. However, the expressed fusion proteins of SiaD$^{D155A}$ and SiaD$^{K250A-D254A}$ were present in the form of inclusion bodies. Therefore, only two mutant proteins (SiaD$^{D138A}$ and SiaD$^{E181A}$) were purified. HPLC assay results showed that the D138A and E181A mutations significantly impaired the DGC activity of SiaD (*Figure 2E*). Together, these results show that the residues involved in GTP and Mg$^{2+}$ binding are essential for SiaD activity.

In contrast to the A-site, c-di-GMP product binding/inhibitory site (I-site) is located at the other end of the GGDEF domain, far away from the A-site (*Figure 3A*, *Figure 2—figure supplement 2*). The I-site could bind an intercalated c-di-GMP dimer in a pocket, which consists of residues from both of the GGDEF domains that are highly conserved among the DGC family (*Figure 2—figure supplement 1*; *Wassmann et al., 2007*). As for SiaD, each of the I-sites consists of residues located at α3 (R201, P197, and V198), a loop between α3 and β3 (D194, T190, N189, L187), and loop between α2 and β2 (D173, Y172, R170, L169) (*Figure 3B*). Size exclusion chromatography (SEC) showed that

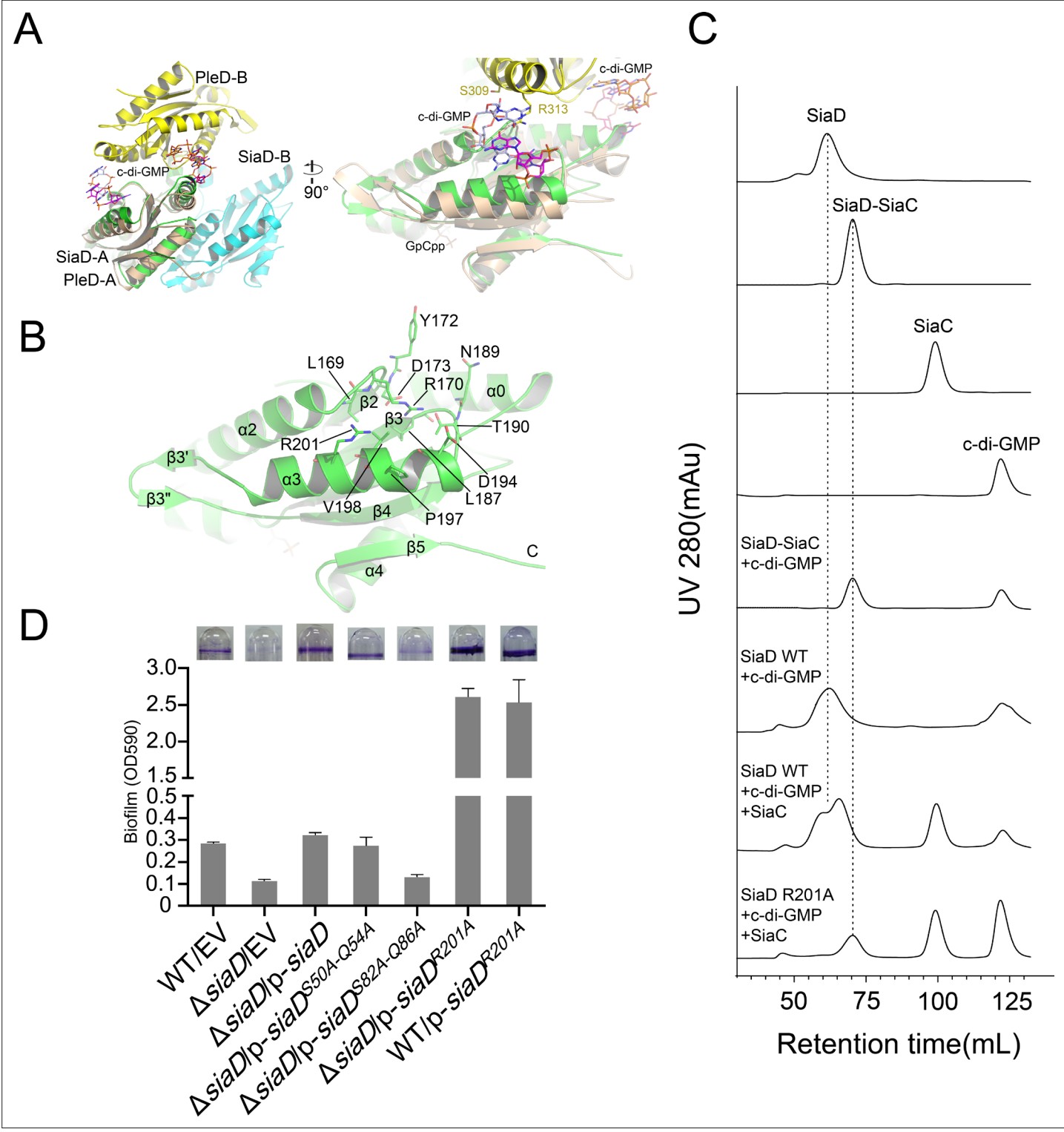

**Figure 3.** Feedback inhibition of I-site of SiaD. (**A**) Superposition of I-sites between SiaD and PleD dimer. The PleD dimer were colored in yellow and orange, and four c-di-GMP molecules observed in PleD structure were colored in ligtblue, magenta, orange, and red, respectively. (**B**) Overview of SiaD I-site. Key residues involved in c-di-GMP binding were show in sticks and labeled. (**C**) SEC analysis of SiaD, SiaD$^{R201A}$ and SiaC–SiaD complex after incubation with c-di-GMP. The retention time of SiaD and the complex was labeled by dotted line. (**D**) SiaD$^{S82A-Q86A}$ mutation weakened the function of SiaC–SiaD complex during biofilm formation, while SiaD$^{R201A}$ basically eliminated the feedback inhibition. The biofilm formation of the indicated strains was displayed with crystal violet staining (up) and quantified with optical density measurement (down).

The online version of this article includes the following figure supplement(s) for figure 3:

*Figure 3 continued on next page*

*Figure 3 continued*

**Figure supplement 1.** Production of c-di-GMP by the indicated protein samples determined by HPLC.

**Figure supplement 2.** Effect of c-di-GMP on SiaD protein thermal stability.

binding of c-di-GMP to the inactive SiaD protein precludes the formation of the 2:4 stoichiometric SiaC–SiaD complex, leading to disordered aggregation (*Figure 3C*). Additionally, the enzyme activity of SiaD was inhibited by c-di-GMP more extensively than the SiaC–SiaD complex (*Figure 3—figure supplement 1*). SiaD appeared to exist as an oligomer in SEC, as the retention time was shorter than that of the SiaC–SiaD complex. Subsequently, mutation of the primary I-site (R201A) restored the SiaC–SiaD interaction (*Figure 3C*) and promoted the DGC activity of SiaD (*Figure 3—figure supplement 1*). Consistently, overexpression of the R201A mutation significantly promoted biofilm formation compared to that of SiaD (*Figure 3D*). In the active state (SiaC–SiaD complex), the addition of excess c-di-GMP did not alter the retention time or peak appearance of the SiaC–SiaD complex (*Figure 3C*), suggesting that the SiaC–SiaD complex retains its 2:4 conformation. However, addition of c-di-GMP inhibited, albeit moderately, activity of the SiaC–SiaD complex (*Figure 3—figure supplement 1*). As expected, this inhibition was eliminated by the R201A mutation (*Figure 3—figure supplement 1*). In addition, the protein thermal shift assay (PTSA) showed that the binding of c-di-GMP to the I-site of the DGC domain promotes protein stability of SiaD, whereas the binding modes for SiaD and SiaC–SiaD complex may be different (*Figure 3—figure supplement 2*). Thus, both the active and inactive forms of SiaD are inhibited by c-di-GMP, suggesting two distinct inhibitory modes for active and inactive SiaD.

## SiaC binds to the stalk domain of SiaD and activates the catalysis of SiaD

The small N-terminal domain and GGDEF domain of SiaD are linked by a long helical stalk domain (α–1). Four SiaC molecules bind to the SiaD stalk, stabilizing the extended shape of the stalk. Upon binding, SiaC does not undergo a significant conformational change compared to its native structure, with an average root-mean-square deviation (r.m.s.d.) of 0.37 Å (*Figure 4—figure supplement 1*). SiaC consists of six antiparallel β-strands and three parallel α-helices. Its interaction with SiaD is mediated predominantly through its α2′ and α3′ helices (secondary elements and residues of SiaD and its symmetry related elements and residues are unlabeled or labeled with an asterisk '*'; the secondary elements and residues of SiaC and its symmetry related elements and residues are labeled with primes "'" or """, respectively), which covers an average of 1764 Å$^2$ of the SiaC protein surface out of a total area of 6036 Å$^2$. The two SiaC molecules proximal to the SiaD DGC domain form a parallel interaction of their α3′ helices with the α–1 stalk domain of SiaD (*Figure 4A*). The side chains of SiaC residues D79′ of the α2′ helix, and R103′, E110′, and D114′ of the α3′ helix form hydrogen bonds with the side chains of SiaD residues R74, D91, R84, R80, K77, and R74 on the stalk helix, respectively. Furthermore, hydrophobic interactions of the SiaC residues L107′ of the α3′ helix, M75′ of the α2′ helix, Q10" located between the β1" and β2" strands, and D29" and Y31" located between the β3" strand and α1" helix with SiaD residues M88, I81, Y85, and R90 of the stalk helix further stabilize the binding of SiaC to the binding motif. In addition, the side chain of the key residue T68′ in the N-terminal portion of the SiaC α2′ helix inserts into a space between the parallel SiaD stalk helices, forming strong hydrogen bonds with the side chains of SiaD residues S82 and Q86 (*Figure 4B* and *Figure 4—figure supplement 2*). Moreover, the SiaC residues Y65′, S69′, and N67′, which are located near T68′, undergo further hydrophilic interactions with SiaD Q86 and D83, while I71′ of SiaC forms a hydrophobic interaction with SiaD V78. Triple mutagenesis of N67′, T68′, and S69′ to alanine or phosphorylation of T68′ significantly prevents the binding of SiaC to SiaD (*Figures 1C and 4C*).

Strikingly, the binding modes of the proximal and distal SiaC dimers are similar due to the highly conserved motif sequences of SiaD (*Figure 4D–F*), except that the corresponding residue R84 from the motif near the DGC domain corresponds to G52 in the other motif, abolishing its hydrophilic interaction with E110′ of SiaC at this site. Nevertheless, these highly conserved interactions between two SiaC-binding motifs suggest an identical binding pattern for the four SiaC monomers to the SiaD dimer, with a K$_d$ value of 49.8 nM (*Figure 4G*). To evaluate the role of the two SiaC-binding motifs during SiaD activation, the SiaD$^{S50A-Q54A}$ and SiaD$^{S82A-Q86A}$ mutants were purified, and SEC and size-exclusion

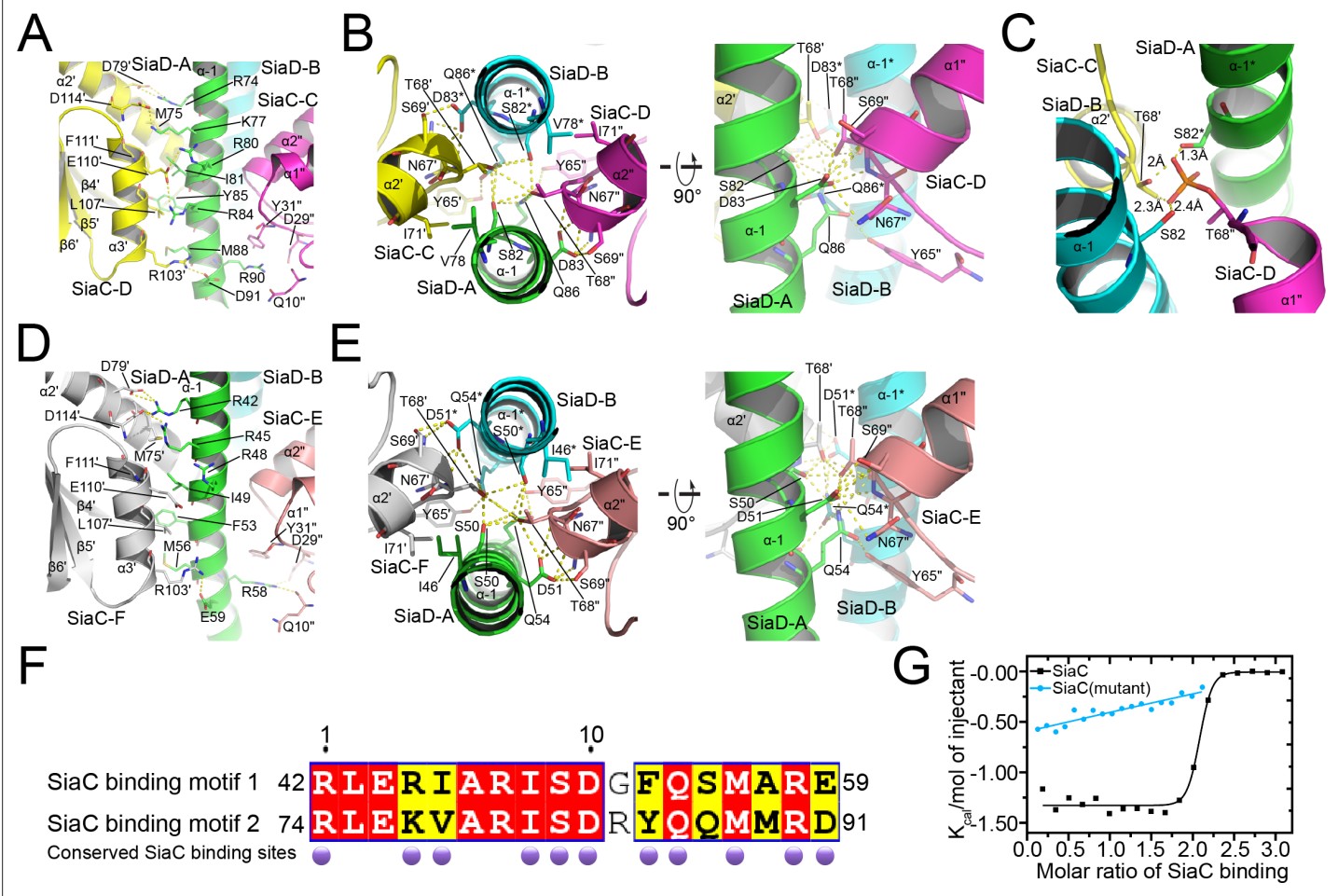

**Figure 4.** Details of SiaC–SiaD interaction. (**A**, **D**) The interactions of SiaC α2′ and α3′ helixes with SiaD α–1 stalk helixes. The secondary elements and residues of SiaD and its symmetry related elements and residues are labeled or with an asterisk; The secondary elements and residues of SiaC and its symmetry related elements and residues are labeled with primes "′" or "″"; (**B**, **E**) the interactions of SiaC key residues N67′, T68′, and S69′ with SiaD α–1 stalk helixes. (**C**) The phosphorylation model of SiaC T68′ abolishes its binding to SiaD by introducing clashes. (**F**) Sequence alignment of the two SiaC binding motifs of SiaD. The identical residues and similar residues between the two motifs are shadowed in red and yellow. Conserved residues involved in the interactions with SiaC are labeled with darkpurple disks. (**G**) Results of isothermal titration calorimetry (ITC) for SiaC (black) or SiaC N67A/T68A/S69A mutant (cyan) binding to SiaD. Plots of molar enthalpy change against SiaC–SiaD or SiaC mutant-SiaD molar ratio are shown.

The online version of this article includes the following figure supplement(s) for figure 4:

**Figure supplement 1.** Superposition of SiaC structure with four SiaC molecules from the SiaC–SiaD complex.

**Figure supplement 2.** The stereo view of omit (fofc) electron density of the key residues N67, T68, and S69 of SiaC at 3.0 σ.

**Figure supplement 3.** Multiple structural alignment and superposition of SiaC with the CheY homolog.

**Figure supplement 4.** Multiple structural alignment and superposition of SiaC with the STAS domains of stressosome homologs.

chromatography coupled with multi-angle light scattering (SEC-MALS) were performed subsequently. For SEC-MALS, purified proteins (2 mg/ml for the SiaC–SiaD complex, 1.5 mg/ml for SiaC, 8 mg/ml for SiaD, 2 mg/ml for SiaC–SiaD$^{S82A-Q86A}$ complex, 2 mg/ml for the SiaC–SiaD$^{S82A-Q86A}$ complex) were injected into a Wyatt Technology WTC-030S5 column, and the UV absorption of protein was monitored. Subsequent data analysis showed that the SiaC–SiaD complex was a dimer (averaged MW = 118.8 kD, theoretical MW = 120.8 kD) and SiaC was a monomer (averaged MW = 14.6 kD, theoretical MW = 14.5 kD). However, the corresponding average molecular weight of the peak fraction for SiaD was 160.5 kD, indicating a pentamer conformation of SiaD in solution, as the theoretical molecular weight of a SiaD pentamer is 157.3 kD (which matched the retention time of SEC) (*Figure 5—figure supplement 1* and *Figure 3C*). Due to the high affinity, both SiaC site mutants (SiaD$^{S50A-Q54A}$ and SiaD$^{S82A-Q86A}$) were still able to form 2:4 stoichiometric complex with SiaC (*Figure 5A–C*). However,

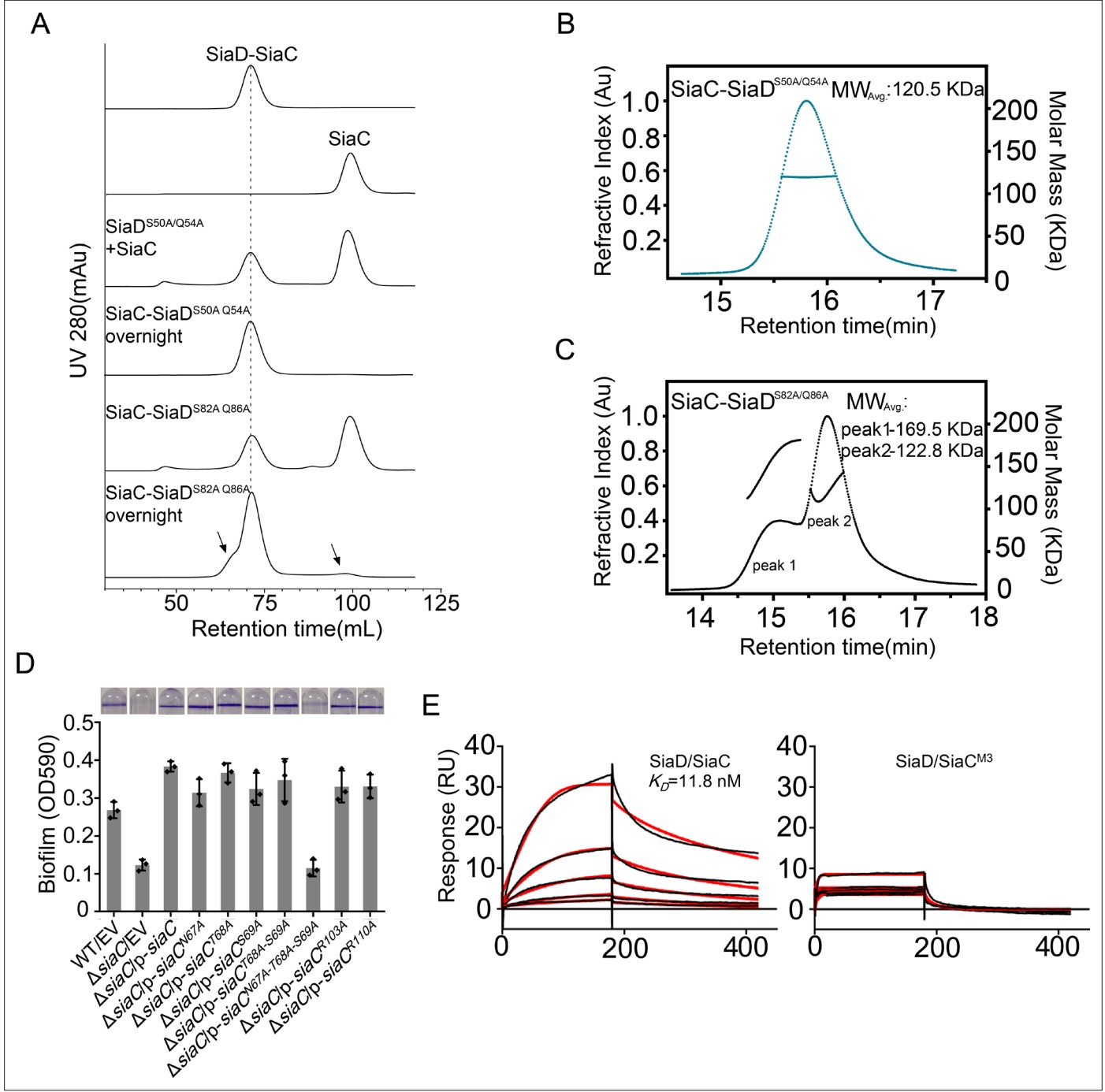

**Figure 5.** Mutation of SiaC–SiaD interface. (**A**) SEC analysis of SiaC-SiaD$^{S50A/Q54A}$ and SiaC-SiaD$^{S82A/Q86A}$ complex. After overnight placement, part of the SiaC-SiaD$^{S82A/Q86A}$ dissociated. The dissociation peaks were marked by black arrow and the retention time of SiaC-SiaD complex was marked by dotted line. (**B, C**) SEC-MALS analysis of SiaC-SiaD$^{S50A/Q54A}$ and SiaC-SiaD$^{S82A/Q86A}$ complex. The protein was separated using a Wyatt Technology WTC-030S5 column. The running buffer contains 20 mM HEPES (pH 7.0), 150 mM NaCl and 1 mM DTT. The linear curve indicates the calculated molecular masses of the samples throughout the peaks. (**D**) N67A/T68A/S69A triple mutation abolished the function of SiaC during biofilm formation. The biofilm formation of the indicated strains were displayed with crystal violet staining (up) and quantified with optical density measurement (down). (**E**) SiaC$^{N67A/T68A/S69A}$ triple mutation (SiaC$^{M3}$) was unable to interact with SiaD. SPR measurements of SiaC or SiaC$^{M3}$ binding at varying concentrations to SiaD. SiaC-His specifically interacted with SiaD with a $K_d$ of 11.8 nM. Shown are measured binding responses (black) and curve fits to a 2:1 interaction (red). Plots are representative from two experiments with similar results. RU, response units; $K_d$, dissociation constant.

The online version of this article includes the following figure supplement(s) for figure 5:

**Figure supplement 1.** SEC-MALS analysis of SiaC–SiaD complex, SiaC and SiaD protein under 2 mg/ml, 2 mg/ml, and 8 mg/ml.

**Figure supplement 2.** Synchrotron solution small angle X-ray Scattering (SAXS) measurements of SiaC–SiaD complex.

part of the SiaC–SiaD$^{S82A-Q86A}$ complex dissociated after overnight incubation, resulting in two peaks in the SEC-MALS results; this was also observed with SEC (*Figure 5A,C*). Consistently, SiaD$^{S82A-Q86A}$ activity was decreased compared to that of SiaD (*Figure 3—figure supplement 1*). Furthermore, overexpression of SiaD$^{S82A-Q86A}$ in the Δ*siaD* mutant failed to restore biofilm formation (*Figure 3D*). These results suggest that mutation of this SiaC-binding motif has a greater influence on the stability and function of the SiaC–SiaD complex. However, neither of the two SiaC-binding motif mutations could fully disrupt SiaC binding, it is difficult to clearly distinguish their role during SiaD activation.

The role of several SiaC residues (N67A, T68A, S69A, R103A, and E110A) in the formation of SiaC–SiaD complex was also evaluated using a quantitative biofilm assay. Although all single mutations and the T68A/S69A double mutation restored biofilm formation in Δ*siaC*, the triple mutant N67A/T68A/S69A failed to restore biofilm formation in Δ*siaC* (*Figure 5D*). This result suggested that N67, T68, and S69 play a key role in the SiaC–SiaD interaction. In addition, surface plasmon resonance and GST pull-down assays confirmed that the triple mutant N67A/T68A/S69A prevents binding of SiaC to SiaD (*Figures 1C and 5E*). Furthermore, in vitro enzymatic analysis showed that the N67A/T68A/S69A mutant drastically impaired the activation of SiaD DGC activity (*Figure 2E*).

## SAXS analyses confirm the oligomerization state of SiaC–SiaD complex

The accuracy of SiaC–SiaD complex conformation obtained from the structure was further confirmed using the synchrotron-based solution small angle X-ray scattering (SAXS). Guinier analysis indicated that the complex radius of gyration (Rg) was 44.54 ± 0.17 Å and the calculated model intensity from the crystal structure fit well to the experimental SAXS data, with a discrepancy $\chi^2$ value of 1.075 in the experimental scattering profiles (*Figure 5—figure supplement 2A* and *Supplementary file 1d*). This finding suggested consistency between the experimental scattering intensity and the crystal structure model, although the models varied in terms of their agreement with the experimental data due to the presence of multiple domains. Moreover, both the experimental pair distance distribution functions (PDDF) and the parabolic appearance of the dimensionless Kratky plots showed an asymmetric peak pattern, indicating that the complex adopts a non-spherical and multiple-domain conformation in solution (*Figure 5—figure supplement 2B*, *C*). The calculated maximum particle dimension (Dmax) is consistent with the longest dimension of the crystal structure (134.8 Å vs. 147.1 Å), and the low-resolution particle model reconstructed from SAXS experimental profiles fits well with the crystal structure, confirming the complex conformation determined from the structure (*Figure 5—figure supplement 2D*).

## SiaC regulates SiaD activity through a dynamic regulatory mechanism

In contrast to the SiaC–SiaD complex, which displays high enzymatic activity, SiaD alone shows little enzymatic activity. Interestingly, the isothermal titration calorimetry assay revealed comparable binding affinities for GTP to SiaD alone (K$_d$ = 12 μM) and the SiaC–SiaD complex, indicating that the binding of SiaC does not promote or interfere with the binding of GTP (*Figure 2D*). Switching between distinct oligomeric states contributes to the activation of several DGCs, such as WspR in *P. aeruginosa* (*Hickman et al., 2005*). The pentamer state of SiaD was determined by SEC-MALS even at low concentration (4 mg/ml), indicating a high consistency of SiaD among various concentrations (*Figure 6A* and *Figure 5—figure supplement 1C*). Therefore, we speculated that SiaC may activate SiaD by switching its oligomeric state. Moreover, to assess the role of the N-terminal domain in the SiaD pentamer, GST pull down was performed. SiaD-Flag, but not SiaD$^{ΔN95}$-Flag, co-immunoprecipitated with GST-SiaD (*Figure 6—figure supplement 1*). Therefore, the N-terminal sequence of SiaD is critical for pentamer formation.

Subsequently, SEC-SAXS experiments were further performed to confirm the above observations. Consistent with the SEC-MALS results, the calculated model intensity of a SiaD pentamer predicted by the GalaxyHomomer server fitted well to the experimental SAXS data, with a discrepancy $\chi^2$ value of 1.116 (*Figure 6B*). The ab initio data showed good consistency (normalized standard deviation = 1.486 ± 0.186) and improved ensemble resolution of 65 ± 5 Å with no symmetry added in reconstruction. By contrast, the experimental PDDF and the parabolic dimensionless Kratky plots for SiaD exhibited a nearly perfect symmetric peak, indicating that SiaD adopts a spherical and single domain conformation in solution (*Figure 6C,D*). Furthermore, the SiaD radius of gyration (Rg = 41.53 ± 0.14 Å) and the maximum particle dimension (Dmax = 127.8 Å) based on SAXS data are similar to those of the

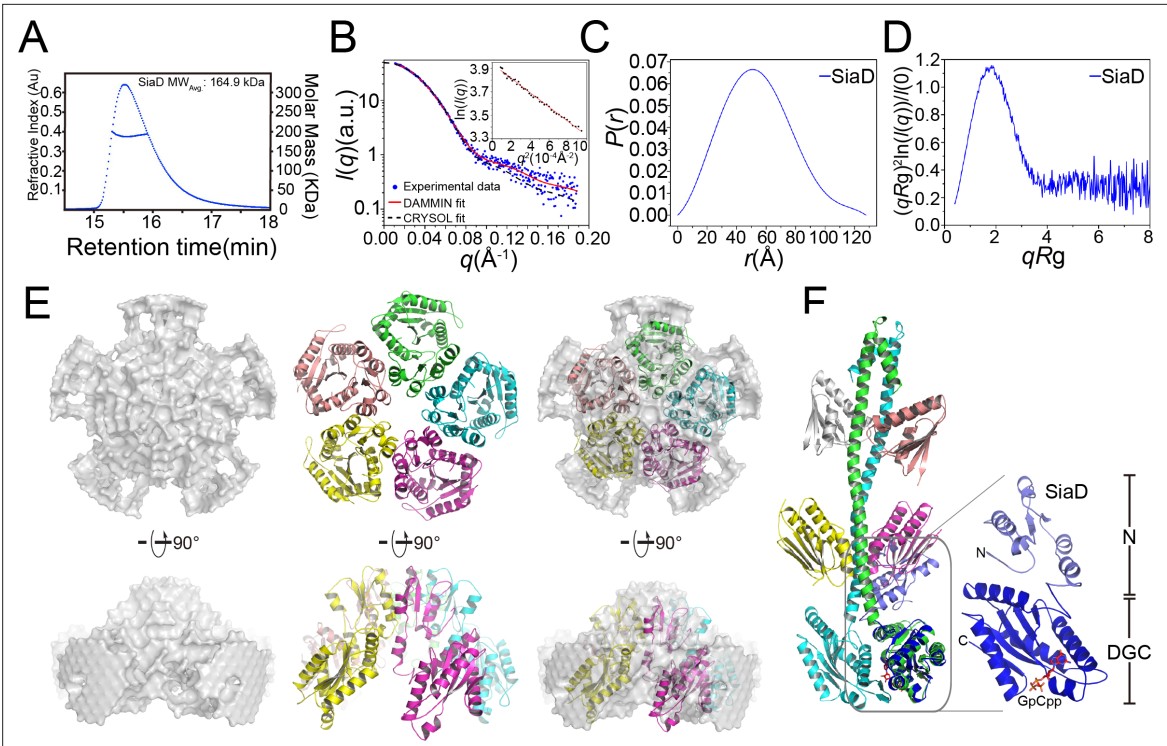

**Figure 6.** SEC-MALS and SAXS measurements of SiaD. (**A**) SEC-MALS analysis of SiaD protein under 4 mg/ml. The protein was separated using a Wyatt Technology WTC-030S5 column. The running buffer contains 20 mM HEPES (pH 7.0), 150 mM NaCl, and 1 mM DTT. The linear curve indicates the calculated molecular masses of SiaD throughout the peaks. (**B–E**) SAXS analysis of SiaD protein. Scattering profile (**B**), PDDFs (**C**), and dimensionless Kratky plots (**D**) of SiaD. The inset in (**B**) is the guinier region with fitting line of the scattering profile. The back-calculated scattering profile of the ab initio model (red line) and atomic model (black dash line) was fitted to the experimental SAXS data of SiaD (dot line). (**E**) SAXS modeling: low-resolution envelope for SiaD protein was shown as a particle model and was superposed with the SiaD model pentamer. (**F**) Modeled structure of SiaD monomer. Superposition of SiaD from SiaC–SiaD complex with the modeled SiaD monomer were shown. The N-terminal domain and C-terminal DGC domain of SiaD are labeled.

The online version of this article includes the following figure supplement(s) for figure 6:

**Figure supplement 1.** The N-terminal stalk is essencial for SiaD–SiaD interaction.

**Figure supplement 2.** Blue native PAGE for SiaD and SiaC–SiaD complex.

**Figure supplement 3.** Analysis of secondary structures of SiaD by cirular dichroism spectrum.

**Figure supplement 4.** Feedback inhibition model of SiaC–SiaD complex.

complex (44.54 ± 0.17 Å and 134.8 Å, respectively), indicating a larger SiaD oligomer than a dimer (*Supplementary file 1d*). The estimated molecular weight of the SiaD oligomer in solution is 159.5 kD, which is near the theoretical molecular weight of the SiaD pentamer (157.3 kD). Additionally, the pentamer oligomer conformation of SiaD alone was confirmed by native polyacrylamide gel electrophoresis (*Figure 6—figure supplement 2*). Furthermore, a low-resolution particle model was reconstructed from the SAXS experimental profiles. The model showed a circular dish-like shape, with five SiaD monomers well fitted into the particle model (*Figure 6E*). Notably, the particle model does not accommodate a model with an extended stalk conformation from the complex, as it is too long. Alternatively, SiaD adopts a compact N-terminal conformation, with the extended stalk domain reduced to five short helices and two β-strands (*Figure 6F*). Then, the secondary structure of SiaD in its active or inactive state was analyzed by far-UV circular dichroism (CD) spectroscopy. To characterize the variations in the secondary structure for SiaD, we tracked CD signals by gradually increasing the SiaC concentration. The CD data suggested that the α-helix increased, while the beta-sheet decreased during SiaD activation (*Figure 6—figure supplement 3* and *Supplementary file 1e*), which supports the model showing that inactive SiaD adopts a compact N-terminal conformation. These biochemical and biophysical results suggest that SiaD may undergo three steps of conformational change upon

SiaC binding and activation: pentamer deoligomerization, reconstruction of the SiaD N-terminal stalk domain, and SiaD dimerization with SiaC binding.

In conclusion, our results redefine the SiaD operon and reveal the dynamic regulatory mechanism of SiaD by SiaC in *P. aeruginosa*. Dynamic switching between the inactive pentamer and active dimer is regulated by SiaC: SiaC induces deoligomerization of the inactive SiaD pentamer and binds to the two conserved motifs in the stalk domain of the reconstructed active SiaD dimer to stabilize its conformation for further catalysis. Phosphorylation of SiaC T68 by the kinase SiaB prevents this binding and thereby shuts down the enzymatic activity of SiaD (*Chen et al., 2020*).

## Discussion

We have shown that SiaC stimulates the DGC activity of SiaD through direct interaction (*Figure 1*). Here, we discovered that SiaD alone has an inactive pentamer conformation, whereas binding of SiaC to its stalk region promotes the formation of an active dimer. DGC catalyzes the condensation of two identical substrate molecules of GTP to form the twofold symmetric c-di-GMP product. As the DGC domain binds only one substrate, DGC activity relies on an oligomerization event that brings two GGDEF domains into close proximity. The formation of distinct oligomeric states contributes to both the activation and auto-inhibition of DGCs. Oligomerization is involved in the modulation of WspR activity through the switch from an active to a product-inhibited dimer via tetrameric assembly (*De et al., 2008*). The activity of WspR is modulated by a helical stalk motif, which is a key regulatory element for oligomerization activation and auto-inhibition (*De et al., 2008*). Similarly, our SiaC–SiaD complex structure contained a long helical stalk of SiaD, which is essential for SiaD modulation through the SiaC–SiaD interaction and SiaD dimer formation. Additionally, *Paul et al., 2007* reported that a dimer conformation represents the enzymatically active state of PleD. Modulation of its active aspartate residue with the phosphoryl analog $BeF_3^-$ leads to stabilization of the PleD dimer, which is the catalytically active form (*Wassmann et al., 2007*). Several lines of evidence have indicated that SiaD has a pentamer conformation. Upon SiaC binding, the compact N-terminus turns into a extended stalk domain and a SiaD dimer forms (*Figure 6F*, *Figure 6—figure supplement 3*). Furthermore, the in vitro assay showed that SiaD alone exhibited negligible DGC activity, but such activity was drastically promoted upon SiaC binding, indicating that switching to the dimer conformation with the aid of SiaC is critical for SiaD activation.

Conformation-induced activation has also been observed in other diguanylate cyclase families, such as the LapD-GcbC system in *Pseudomonas,* where sensing and binding to c-di-GMP leads to conformational changes that trigger the signal transduction across the membrane (*Dahlstrom et al., 2015*; *Giacalone et al., 2018*). Moreover, the critical conformation changes are predominantly supported and regulated by the long coiled coil regions for stabilization of the GGDEF domain homodimers, such as Zn-dependent activation of DgcZ (*Zähringer et al., 2013*) and the red light activated bacteriophytochrome diguanylyl cyclase PadC (*Gourinchas et al., 2017*). However, the $SiaD_5$ to $SiaD_2$-$SiaC_4$ conformation switch, as well as the stabilization of the long stalk region by four SiaC, redefines this family. Two SiaC binding motifs are located at the N-terminal stalk of SiaD. It is possible that SiaC binds sites in a sequential manner. However, the amino acid sequences of the two SiaC-binding motif are highly conserved, and the ITC experiments showed a continuous 'S' curve for SiaC–SiaD binding, indicating that there was no significant difference in affinity between the two binding sites (*Figure 4G*). Subsequently, our mutagenesis study showed that the two SiaC-binding motif mutations still resulted in the formation of the SiaC–SiaD complex, making it difficult to clarify their individual roles.

Product inhibition represents another layer of DGC modulation (*Chan et al., 2004*; *Christen et al., 2006*; *De et al., 2008*). Product feedback inhibition of DGCs may establish precise cellular threshold concentrations of c-di-GMP available to be read by downstream targets. Meanwhile, the product inhibition allows for a rapid enzymatic response to environmental fluctuations in that the rate of c-di-GMP synthesis can be manipulated without having to translate or degrade a given DGC. Many of the DGCs are subject to non-competitive product inhibition through the binding of c-di-GMP to the allosteric I-site on the surface of the GGDEF domain. In PleD and WspR, an intercalated c-di-GMP dimer binds the primary I-site and a secondary I-site, thereby immobilizing the GGDEF domains in a non-catalytic state (inhibition by domain immobilization) (*Chan et al., 2004*; *Christen et al., 2006*; *De et al., 2008*). For PleD, two modes of product inhibition have been proposed. In inactive PleD, the primary I-site cross-linked to the secondary I-site of the REC2

domain stabilizes an inactive conformation. In active PleD, dimeric c-di-GMP cross-links the primary I-site with a secondary I-site on the neighboring DGC domain to immobilize the active sites away from each other. Analogously, in this study, we revealed two distinct inhibitory modes for SiaD. In the inactive state, c-di-GMP inhibits SiaD activity by preventing the formation of the SiaC–SiaD complex. Since SiaD has a canonical primary I-site and an arginine-rich N-terminus, it is possible that c-di-GMP inhibits SiaD by cross-linking the GGDEF and N-terminal domains to prevent dimerization (activated state) (*Chan et al., 2004*; *Christen et al., 2006*). On the other hand, tge PTSA showed that the binding mode of c-di-GMP may differ between inactive and active SiaD (*Figure 3—figure supplement 2*). In the active state, c-di-GMP did not destabilize the 2:4 stoichiometric conformation, but inhibited DGC activity. The secondary I-site on the GGDEF domain (R125 for SiaD, and R313 for PleD) is conserved among SiaD and PleD (*Figure 2—figure supplement 1*). We deduced that the intercalated c-di-GMP dimer inhibits SiaD activity in a manner analogous to that of PleD (*Wassmann et al., 2007*), and the rotation of the paired GGDEF domains in the SiaC–SiaD complex may be facilitated by the flexible linkers that connect the N-terminal stalk and GGDEF domains (*Figure 6—figure supplement 4*).

Modulation of activity has been reported for several DGCs, including WspR, PleD, and DgcB (*Chan et al., 2004*; *Hickman et al., 2005*; *Meek et al., 2019*). Notably, protein phosphorylation plays a critical role during DGC modulation, as WspR, PleD and DgcB are all activated by phosphorylation (*Hickman et al., 2005*; *Meek et al., 2019*; *Paul et al., 2007*). Consistent with this function, we have shown that protein phosphorylation plays a central role in the modulation of SiaD activity. However, apparent differences in DGC activation can be observed between SiaD and WspR or PleD. First, the phosphorylation site of SiaC (threonine) differs from that of WspR or PleD (aspartate) (*Figure 4—figure supplement 3*). Second, WspR and PleD are activated by phosphorylation, whereas phosphorylation of SiaC prevents the SiaC–SiaD interaction, and thus SiaD activation. Most importantly, SiaD does not contain a N-terminal sensory domain to specifically receive phosphate group from upstream pathways, and therefore modulation of SiaD activity is facilitated by phosphorylation of its binding partner SiaC, with SiaC functioning as a sensory domain of SiaD. The biological significance of the mode of SiaD activation is interesting and confusing. At present, it remains unclear why bacteria encode an extra binding protein for DGC activation. Further investigations are needed to elucidate the biological significance of this phenomenon.

Bacterial PSSs constitute regulatory modules that act through reversible protein-protein interactions and phosphorylation events. PSSs are reportedly involved in modulation of DGC activity. In *P. aeruginosa*, the activity of DGC HsbD was modulated by the HptB-HptR-HsbA PSS (*Valentini et al., 2016*). In addition, the BgrSTUWV PSS modulates the activity of DGC BgrR in *Sinorhizobium meliloti* (*Baena et al., 2017Baena et al., 2017Baena et al., 2017*). The DGC activities of HsbD and BgrR are modulated by their STAS-domain binding partners HsbA and BrgV. We revealed that SiaD is modulated by SiaABC PSS. Interestingly, its binding partner SiaC contains a domain of unknown function, DUF1987, which is structurally related to the STAS domain of HsbA or BrgV (*Figure 4—figure supplement 4*). As the DUF1987 domain is widespread among bacteria, we suggest that a common function of DUF1987 domain-containing proteins is modulating the activity of adjacent proteins.

Overall, we have provided structural details of the mechanism through which SiaC activates SiaD function and thus biofilm development in *P. aeruginosa*. These structural data provide the molecular basis for SiaC–mediated conformation changes of SiaD from the inactive pentamer to active dimer form, which supports the elucidation of the functions of DUF1987 domains in other bacteria. Based on the data from this and previous studies (*Chen et al., 2020*), we propose a model of SiaD modulation (*Figure 7*). To the best of our knowledge, this is a novel mechanism of DGC activation by a binding partner protein. The phosphatase SiaA and kinase SiaB modulate the phosphorylation state of SiaC. Phosphorylation of SiaC precludes the SiaC–SiaD interaction, whereas unphosphorylated SiaC directly interacts with the stalk of SiaD, facilitating its conformational transition from inactive pentamer to active dimer to synthesize c-di-GMP. High levels of c-di-GMP then repress SiaD activity via I-sites through two distinct modes. Through modulation of the DGC activity of SiaD, bacteria can rapidly respond and adapt to environmental changes. Moreover, this unique regulatory model expands our general understanding of GGDEF modulation and provides a new starting point for elucidating the molecular mechanisms responsible for modulation of the DGC family.

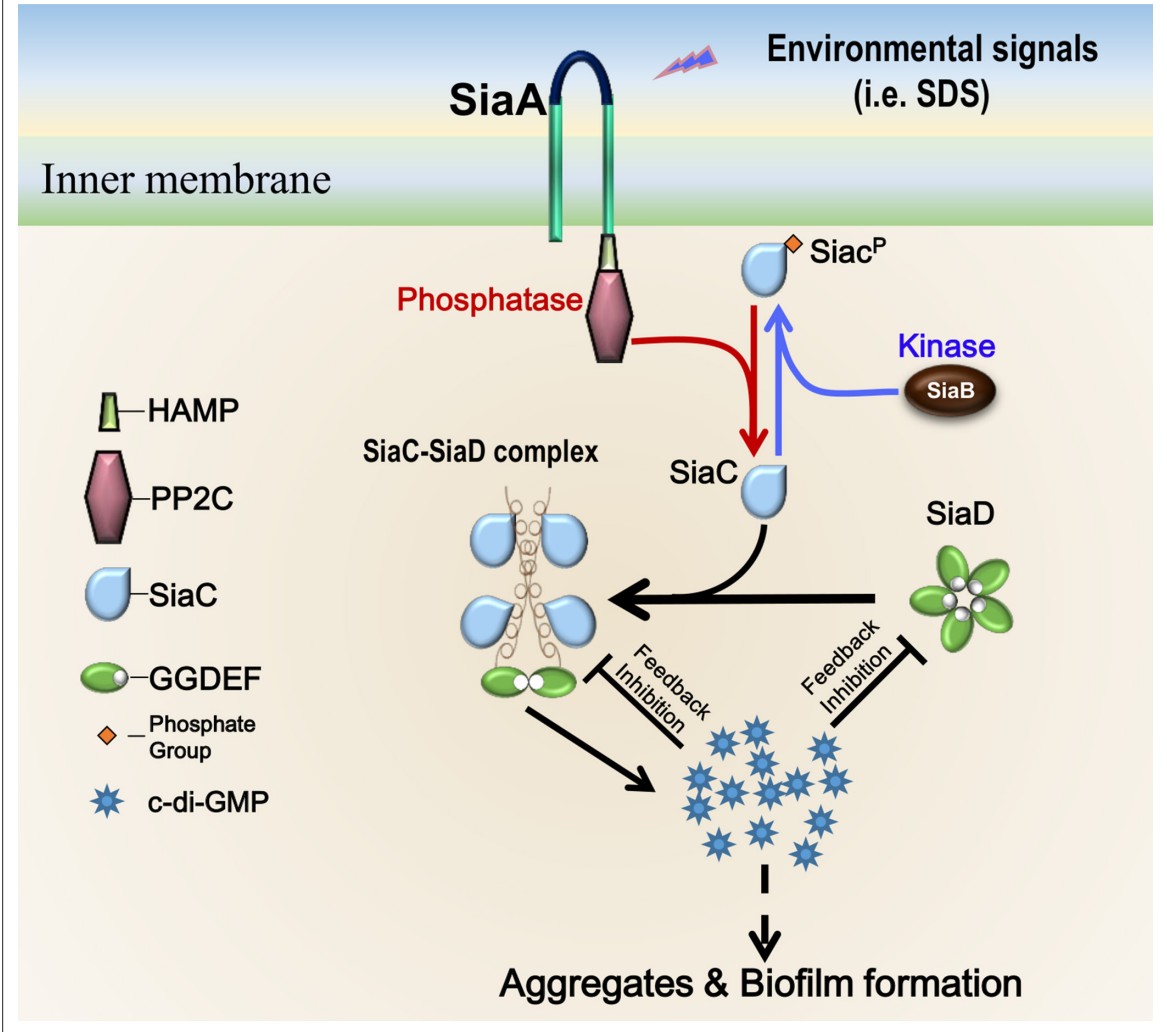

**Figure 7.** Proposed model for SiaC-mediated activation of SiaD. Without SiaC binding (depletion or phosphorylation of SiaC), SiaD alone forms an inactive pentamer conformation in solution. To activate SiaD, four SiaC binds to the stalks of two SiaD and promotes the formation of SiaD dimer. High level c-di-GMP then represses the SiaD activity via I-site through two distinct modes.

## Materials and methods

### Bacterial strains and culture conditions

Bacterial strains, plasmids, and primers used are listed in *Supplementary file 1a,b*. Strains were incubated in LB medium. When needed, appropriated antibiotics were supplemented: 150 μg/ml carbenicillin for *P. aeruginosa* and 100 μg/ml carbenicillin for *E. coli*.

### Biofilm formation assay

Biofilm formation assay was performed as described previously with modification. Briefly, overnight cultures grown in LB medium were diluted (1:1000) into 1 ml fresh LB medium in glass tube and incubated statically at 25 °C for 22 hr. Biofilms attached to the sides of the glass tubes were washed twice gently with sterile water and stained with crystal violet (CV). Quantification of biofilm biomass was performed by dissolution of the CV-stained biofilm with ethanol and the CV solution was measured at an absorbance of $OD_{590}$ nm. The assay was performed at least three times with a minimum of three replicates for each strain.

## Protein expression and purification

DNA fragments encoding full length of SiaD, truncated SiaD$^{\Delta N40}$, SiaD$^{D138A}$, SiaD$^{E181A}$, SiaD$^{R201A}$, SiaD$^{S50A-Q54A}$, SiaD$^{S82A-Q86A}$, SiaC, and SiaC$^{N67A/T68A/S69A}$ mutant were subcloned to pSumo vector and expressed in *E. coli* BL21(DE3) cells, respectively. The cells were grown in 2xYT medium with Kanamycin antibiotics, and 1 mM final concentration of IPTG was added into the medium once OD$_{600}$ reached 0.6. The cells were further grown for 18 hr at 25 °C and then were centrifuged down for further protein purification. The entire purification was carried out at 4 °C. The cell pellets were suspended in buffer A containing 50 mM Tris–HCl, pH 7.4, 500 mM NaCl, 20 mM imidazole and 10 mM MgCl$_2$, and lysed on ice by using French press (Union). The supernatant after the high speed centrifugation (13,000 g for 35 min) were loaded onto Ni-NTA column. The protein in the column is further washed and eluted by using 5%–50% gradient buffer B containing 50 mM Tris–HCl, pH 7.4, 500 mM NaCl, 500 mM imidazole, and 10 mM MgCl$_2$. Subsequently, for SiaD or SiaC protein alone, the N-terminal sumo-His tag attached to the protein were remove by incubating the protein with recombinant ULP1 enzyme at 4 °C for 5 hr; for SiaC–SiaD complex protein, sumo tagged SiaC and SiaD protein were mixed in molar ratio 3:1 and then mixed with ULP1 enzyme for tag cleavage. Lastly, the protein or protein complex were concentrated and loaded to a Superdex200 gel-filtration column with buffer containing 25 mM Tris–HCl, pH 7.5, 150 mM NaCl, and 1 mM DTT for further purification. The protein or protein complex were collected and concentrated for crystallization or assays. For SEC, the protein or protein complex were loaded to Superdex200 gel-filtration column with buffer containing 20 mM HEPES, pH 7.0, 300 mM NaCl, and 1 mM DTT. Among them, SiaD, SiaD$^{R201A}$, and their corresponding complexes were purified with sumo-His tag.

## SiaC–SiaD complex crystallization

The SiaC–SiaD complex was concentrated to 10 mg/ml and incubated with GpCpp (GTP analogue, Jena Bioscience) in molar ratio 1:5. Subsequently, the complex was centrifuged 12,000 g at 4 °C for 1 hr, and set up sitting drops against reservoir solutions by mixing 1 μl protein solution with 1 μl reservoir solution at 18 °C. The thin-plate like crystals appear within 3 weeks under reservoir solution containing 0.01 M Spermidine trihydrochloride and 15% w/v PEG 3350, and were flash frozen in protectant containing reservoir with 20 % glycerol (v/v).

## Data collection and structure determination

The crystal diffraction data were collected at the beamline BL19U1 of the Shanghai Synchrotron Radiation Facility (SSRF) (*Supplementary file 1c*), and processed with HKL3000 (*Minor et al., 2006*). The structure was solved by molecular replacement with the conserved DGC domain structure of SiaD homolog WspR (pdb code: 3I5A) and previously published SiaC structure by our group (pdb code: 6KKP) as search models respectively, and refined by using program suit Phenix (*Adams et al., 2002*). Electron density interpretation and model building were performed using the computer graphics program Coot (*Emsley and Cowtan, 2004*). The final structures were visualized by PyMol software. The atomic coordinate and structure factor are deposited to PDB with accession code 7E6G.

## Isothermal titration calorimetry (ITC) assay

Isothermal titration calorimetry assays were performed by using MicroCal iTC200 (GE Healthcare) or MicroCal PEAQ-ITC (Malvern) at 25 °C. SiaD protein was loaded into the cell, and GTP or SiaC/SiaC mutant was diluted before loading into the syringe. An 0.4 μl initial injection was followed by 19 injections of GTP solution at 150 s intervals. MicroCal PEAQ-ITC Analysis Software (Malvern) was used to analyze the raw ITC data.

## Size-exclusion chromatography multi-angle light scattering (SEC-MALS)

The absolute molar masses of SiaD, SiaC, and SiaC–SiaD complex was assessed by size-exclusion chromatography multi-angle light scattering. 50 μl, 2–8 mg/ml purified protein was injected into HPLC system (Agilent Technologies) and separated by using Wyatt Technology WTC-030S5 column with running buffer containing 20 mM HEPES, pH 7.0, 150 mM NaCl, 10 mM DTT. The corresponding molecular weight of the fraction peaks was evaluated and analyzed by using Wyatt Technology SEC-MALS system, which had been normalized by using 2 mg/ml standard BSA sample. The final observed molecular weight of the protein is shown in *Figures 3 and 5* and *Figure 5—figure supplement 1*.

## Synchrotron solution small angle X-ray scattering (SAXS) measurements

The SAXS data of SiaD or SiaC–SiaD complex were collected at the beamline BL19U2 of the Shanghai Synchrotron Radiation Facility (SSRF). All the parameters for data collection and softwares employed for analysis are summarized in *Supplementary file 1d*.

For data collection and processing, multiple concentrations of SiaD or SiaC–SiaD complex purified from gel-filtration in the solution containing 20 mM HEPES, pH 8.0, 150 mM NaCl, 1 mM DTT were used in the Small angle X-ray scattering measurements at 25 °C. The SAXS data under various concentrations showed highly similarity and the data from 8 mg/ml SiaD or 11 mg/ml SiaC–SiaD complex were further analyzed. The scattered X-ray photons were recorded with a PILATUS 1 M detector (Dectris, Switzerland) at BL19U2. The setups were adjusted to achieve scattering q values of $0.009 < q < 0.45$ Å$^{-1}$, where $q = (4\pi/\lambda)\sin\theta$, and $2\theta$ is the scattering angle. The working energy is 13.43 keV ($\lambda = 0.923$ Å). 150 μl of protein sample was injected, and 1.5 s exposures were collected in succession. 2D scattering images were converted to 1D SAXS curves by the software package BioXTAS RAW (*Hopkins et al., 2017*). Scattering profiles of the proteins were calculated by subtracting the background buffer contribution from the sample-buffer profile. Program PRIMUS was used to perform Guinier analysis of the low q data, which provides an estimate of the radius of gyration (Rg) (*Konarev et al., 2003*). Regularized indirect transforms of the scattering data were carried out with the program GNOM to obtain P(r) functions of interatomic distances (*Svergun, 1992*). The P(r) function has a maximum at the most probable intermolecular distance and goes to zero at Dmax, the maximum intramolecular distance. The values of Dmax were chosen to fit with the experimental data and to have a positive P(r) function. The volume of correlation (Vc) was calculated using the program RAW, and the molecular weights of solutes were calculated on a relative scale using the Rg/Vc power law developed by *Rambo and Tainer, 2013*, independently of protein concentration and with minimal user bias. The theoretical scattering intensity of the atomic structure model was calculated and fitted to the experimental scattering intensity using program CRYSOL (*Svergun et al., 1995*).

For Ab initio shape reconstructions, three-dimensional bead models that fit with the scattering data were built with the program DAMMIN, with data extending up to 0.25 Å$^{-1}$, using slow-mode settings (*Svergun, 1999*). The models resulting from 20 independent runs were superimposed using the DAMAVER suite yielding an initial alignment of structures based on their axes of inertia followed by minimization of the normalized spatial discrepancy (NSD) (*Petoukhov et al., 2012*). The aligned structures were then averaged, giving an effective occupancy to each voxel in the model, and filtered at half-maximal occupancy to produce models of the appropriate volume that were used for all subsequent analyses. For SiaC–SiaD complex, although the experimental data fitted well with the crystal structure (CRYSOL$\chi^2$=1.388), all the models were variety in terms of agreement with the experimental data due to the multiple domains. For SiaD, with no symmetry added in reconstruct, the ab initio has good consistency (NSD = 1.163 ± 0.109) and better ensemble resolution 51 ± 4 Å. When using fivefold symmetry, the influence of the anisometry condition increased the ensemble variability (*Tuukkanen et al., 2016*).

For Homology Modeling using SAXS Data, SiaD pentamer structure was predicted from a monomer sequence using GalaxyHomomer server (*Baek et al., 2017*). It performs both template-based modeling and *ab initio* docking, and adopts additional model refinement that can consistently improve model quality. First, two rounds of operations were performed independently, and ten models were obtained. Second, the models were ranked according to the agreement with the experimental scattering profiles using CRYSOL. Finally, less reliable loop or terminal regions of the best model were re-modeled. The figure was generated by using GraphPad Prism five and PyMol software.

## Circular dichroism

CD spectra were collected using an AppliedPhotophysics Chirascan spectropolarimeter. Protein samples were tested in 10 mM Tris–HCl, pH 7.5 and 500 mM NaCl in 0.1-cm-pathlength quartz cuvettes. The far-UV CD spectrum of protein samples were recorded in the range of 250–200 nm at 25 °C, with 1 s/point scanning speed and 1 nm step. Three biological repeats were set for each sample. Three scans were averaged to obtain the final spectra for each sample. The raw data was processed using the software CDtoolX (*Miles and Wallace, 2018*). The contribution of the buffer was substracted. As SiaC conformation has no significant change in SiaC–SiaD complex compared to its native structure, the CD signal of SiaC control was substracted from that of the corresponding

SiaD–SiaC mixture sample to obtain the CD spectra of SiaD in mixture sample. A web-server BeStSel (Beta Structure Selection) was used for secondary structure determination (*Micsonai et al., 2018*).

### Protein thermal shift assay (PTSA)

Protein thermal shift assay was performed by QuantStudio 6 Flex (Thermo Fisher Scientific). The SiaD, SiaD$^{R201A}$, SiaC, and SiaC–SiaD complex were diluted with PTS buffer to 5 μM and mixed with 5× SYPRO Orange dye (Sigma-Aldrich). After incubation with 25 μm of c-di-GMP, the mixtures were added to a 96-well plate and the temperature program (25°C–90°C) was run. Subsequently, the analysis software Protein Thermal Shift Software 1.3 was used to analyze the melting curve and fit the $T_m$ value.

### Blue-Native PAGE

The blue native PAGE (BN-PAGE) was performed as previously described (*Wittig et al., 2006*). Briefly, 8 % BN-PAGE was cast accordingly. 3× loading buffer (20 % glycerol, 1.05 % Coomassie G250, and 1.5 M 6-aminohexanoic acid) was mixed with SiaD (25 μg) alone or SiaC–SiaD mixture (SiaD mixed with excess SiaC [35 μg]) before electrophoresis. After 50 min electrophoresis at 15 mA, the gel was stained by Coomassie blue R250.

### GST pull-down assay

To test the interaction between SiaD and SiaC, 50 μg purified GST-SiaD was mixed with 15 μl prewashed glutathione magnetic beads on a rotator for 2 hr at 4 °C before washed three times by reaction buffer. To avoid non-specific interaction, the beads was then blocked with 5 % skim milk (m/v) 1 hr at 4 °C before addition of bacterial lysate containing SiaC-Flag or SiaC$^{N67A/T68A/S69A}$-Flag. After six times washing by reaction buffer, samples were subject to western blotting assays. To test the interaction between SiaD and SiaD$^{\Delta N95}$, a similar procedure was used.

### Surface plasmon resonance (SPR) experiments

Surface plasmon resonance (SPR) experiments were performed to measure the binding kinetic parameters between proteins. After the activation of the surface using NHS and EDC (1:1, v/v), the protein was immobilized on a CM5 sensor chip (GE healthcare) by using standard amine-coupling at 25 °C with buffer PBS-P (20 mM phosphate buffer, 2.7 mM NaCl, 137 mM KCl, 0.05% surfactant P-20, pH 7.4). The protein concentration was fixed at 50 ng/μl and the immobilization level of protein was about 5000 RU (response unit, 1 RU response value is roughly equivalent to 1 pg/mm$^2$ change of the concentration of the bound substance on the chip surface). After coupling, unreacted NHS ester groups were blocked with ethanolamine. After equilibration by running buffer (10 mM Tris–HCl, 150 mM NaCl, 0.05 % surfactant P-20, pH 7.5), different concentrations of analytes were serially injected into the channel to evaluate the binding kinetic parameters. A reference channel was only activated and blocked to eliminate compound unspecific binding to the surface of the chip. An extra wash with 2 mM NaCl was added to remove the last remaining sample in the pipeline. The on-rates and the off-rates of the compounds were obtained using the Biacore T200 Evaluation Software (GE Healthcare).

### Determination of C-Di-GMP by HPLC

Synthesis of c-di-GMP were carried out in 50 mM Tris–HCl, pH 7.5, 150 mM NaCl. When needed, 5 mM MgCl$_2$ or MnCl$_2$ and 2 mM EDTA (Ethylene Diamine Tetraacetic Acid) was added. Subsequently, 0.7 μM SiaD (or) alone or 0.7 μM SiaD with 1.4 μM SiaC were add to the reaction mixture at room temperature. Reactions using 0.7 μM SiaD$^{\Delta N40}$ alone or 0.7 μM SiaD$^{\Delta N40}$ with 1.4 μM SiaC were used as a control. To evaluate the allosteric inhibition of c-di-GMP, 50 μM c-di-GMP was added to the reaction mixture. Reactions were initiated by the addition of 50 μM GTP to the mixture and allowed to incubate for different time periods at 37 °C. Reactions were terminated by heating the samples at 95 °C for 5 min. Precipitated proteins were removed by centrifugation after which the supernatant was filtered through a 0.22 mm membrane. of sample was loaded for HPLC (Shimadzu, Japan), with 254 nm as detection wavelength. Symmetry C18 Column (4.6 mm × 25 cm) (Waters) was used with solvent A (10 mM ammonium acetate in water) and solvent B (10 mM ammonium acetate in methanol) at a flow rate of 0.2 ml/min. Eluent gradient is as follows: 0–9 min with 1 % B; 9–14 min with 15 % B; 14–19 min with 25 % B; 19–35 min with 1% B. For SiaD$^{R130G}$, a similar strategy was applied.

## Acknowledgements

This work was supported by the National Natural Science Foundation of China (32170188 and 31870060 to HL, 21722802 and 91853118 to LZ, 31700064 to GC), ShaanXi Science and Technology Innovation Team (2019TD-016). We thank the staffs from beamline BL19U1 and BL19U2 of National Facility for Protein Science in Shanghai (NFPS) at Shanghai Synchrotron Radiation Facility (SSRF) for the assistance and support during crystal diffraction and SAXS data collection and analysis. We thank the staff members of the Large-scale Protein Preparation System at the National Facility for Protein Science in Shanghai (NFPS), Zhangjiang Lab, Shanghai Advanced Research Institute, Chinese Academy of Science, China for providing technical support and assistance in data collection and analysis. We thank Prof. Jianhua Gan from School of life science, Fudan University, for the assistance during structure determination.

## Additional information

### Funding

| Funder | Grant reference number | Author |
| --- | --- | --- |
| National Natural Science Foundation of China | 32170188 | Haihua Liang |
| National Natural Science Foundation of China | 31870060 | Haihua Liang |
| National Natural Science Foundation of China | 21722802 | Liang Zhang |
| National Natural Science Foundation of China | 91853118 | Liang Zhang |
| National Natural Science Foundation of China | 31700064 | Gukui Chen |
| ShaanXi Science and Technology Innovation Team | 2019TD-016 | Haihua Liang |

The funders had no role in study design, data collection and interpretation, or the decision to submit the work for publication.

### Author contributions

Gukui Chen, Conceptualization, Data curation, Formal analysis, Funding acquisition, Investigation, Methodology, Project administration, Resources, Software, Supervision, Validation, Visualization, Writing – original draft, Writing – review and editing; Jiashen Zhou, Formal analysis, Methodology, Software, Validation, Writing – original draft; Yili Zuo, Data curation, Formal analysis, Resources, Software, Validation; Weiping Huo, Formal analysis, Investigation, Methodology, Software; Juan Peng, Meng Li, Formal analysis, Software; Yani Zhang, Writing – review and editing; Tietao Wang, Software; Lin Zhang, Methodology; Liang Zhang, Funding acquisition, Methodology, Supervision, Writing – original draft, Writing – review and editing; Haihua Liang, Funding acquisition, Writing – review and editing

### Author ORCIDs

Tietao Wang http://orcid.org/0000-0001-8540-436X
Liang Zhang http://orcid.org/0000-0002-7672-1168
Haihua Liang http://orcid.org/0000-0001-9639-1867

### Decision letter and Author response

Decision letter https://doi.org/10.7554/67289.sa1
Author response https://doi.org/10.7554/67289.sa2

## Additional files

### Supplementary files

• Supplementary file 1. Supplementary tables. (a) Strains and plasmids used in this study. (b) Primers used in this study. (c) Data collection and refinement statistics. (d) Data collection and refinement statistics. (e) Estimated secondary structure content of SiaD.

• Transparent reporting form

• Source data 1. Source data for figures.

### Data availability

Diffraction data have been deposited in PDB under the accession code 7E6G.

The following dataset was generated:

| Author(s) | Year | Dataset title | Dataset URL | Database and Identifier |
|---|---|---|---|---|
| Zhou JS, Zhang L, Zhang L | 2021 | Crystal structure of diguanylate cyclase SiaD in complex with its activator SiaC from Pseudomonas aeruginosa | https://www.rcsb.org/structure/7E6G | RCSB Protein Data Bank, 7E6G |

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
