## [Decision Letter]

**Acceptance summary:**

The study provides novel mechanistic insights into the activation of a diguanylate cyclase, an enzyme that controls bacterial biofilm formation, by its binding partner protein. The work also assigns a molecular function to DUF1987 domains that also occur in other bacterial proteins, potentially suggesting a similar mode of partner-protein regulation.

**Decision letter after peer review:**

Thank you for submitting your article "Structural basis for diguanylate cyclase activation by its binding partner in *Pseudomonas aeruginosa*" for consideration by *eLife*. Your article has been reviewed by 3 peer reviewers, including Holger Sondermann as Reviewing Editor and Reviewer #1, and the evaluation has been overseen by Gisela Storz as the Senior Editor.

Based on an illuminating structure-function study, a novel mechanism is uncovered for the activation of a bacterial diguanylate cyclase that controls biofilm formation in *Pseudomonas aeruginosa* via the production of the second messenger c-di-GMP. By providing structural models for activated and autoinhibited states of the enzyme and revealing how the enzyme transitions between them through the action of a binding partner, a convincing model is put forward that could pave the way for comparative studies on proteins with similar folds and regulation.

The structure determinations by X-ray crystallography and small-angle X-ray scattering were thorough and their interpretation convincing. Quantitative binding, enzyme activity and biofilm assays provide further validation of the proposed model and are consistent with the structural analysis. Although the reviewers were enthusiastic about the study that provides first insight into a novel concept of diguanylate cyclase activation, a few key areas were identified, which are crucial aspects of the activation model and for which further support would strengthen the mechanistic arguments. Thus, the reviewers felt unanimously that a deeper characterization of the following points would be within the scope of the present study and were viewed as being essential.

Essential revisions:

1. A central feature of the activation model is the stabilization of a SiaD dimer by four molecules of SiaC. A mutagenesis study of the two individual SiaC binding sites on SiaD would likely provide insight into the cooperativity of the activation process and may reveal an order of events or relative importance of the binding sites. As readout, enzyme activity (c-di-GMP production) could be correlated with complex formation (via multi-angle laser light scattering). Please also comment whether SiaC is monomeric or dimeric in solution in the absence of SiaD.

2. A particular regulatory feature of a subset of diguanylate cyclases, including SiaD, is product inhibition via a distinct c-di-GMP binding site. The authors identified such a site on SiaD and demonstrate that c-di-GMP inhibits SiaD (albeit at fairly high second messenger concentration). Considering the central role of inhibitory-site regulation in diguanylate cyclase regulation and c-di-GMP signaling networks, a more thorough analysis of this site on SiaD would be warranted. Minimally, the impact of inhibitory-site mutations on *Pseudomonas* biofilm formation, SiaD activation and SiaD-SiaC complex stability should be assessed. A central question is whether c-di-GMP binding to the allosteric, inhibitory site destabilizes the SiaC-SiaD complex.

3. Central to the model of SiaD activation is the restructuring of the N-terminal domain of the enzyme, from an apparent globular fold containing β-strands to a long, extended helix. However, data supporting this transition is sparse. One way to potentially clarify this mechanism is to perform Circular Dichroism (CD) to determine the secondary structure and folding properties of SiaD in the presence and absence of SiaC.

Essential textual changes were summarized elsewhere.

Addressing these points would substantially strengthen the conclusion of the study, providing a more complete model of SiaD regulation and bacterial biofilm formation.

*Reviewer #1:*

Signaling via the bacterial second messenger c-di-GMP has emerged as a major pathway to control cellular physiology, foremost in the transition from a free-swimming to a sessile, social lifestyle. The production of c-di-GMP in bacteria is catalyzed by GGDEF domain-containing enzymes that act as diguanylate cyclases. Their activity is regulated by various mechanisms, but for many we lack structural insight of how this activity modulation is achieved. The manuscript by Chen et al. describes an interesting molecular mechanism responsible for the activation of a diguanylate cyclase in *P. aeruginosa*, SiaD, by its binding partner protein, SiaC. Based on structural analyses of a catalytically active SiaC-SiaD complex bound to a GTP analog and an autoinhibited SiaD, the authors deduce a mechanism by which four SiaC proteins stabilize an elongated, dimeric conformation of SiaD, positioning the GGDEF domains within the dimer for catalysis. In contrast, the inactive state appears to be a homo-pentamer, with each SiaD protomer adopting a more globular fold. Binding studies and biofilm assays involving structure-guided mutant variants of the proteins further support the proposed mechanism.

The observed protein-protein interaction via a protein of unknown function, the DUF1987 domain, may be a diverse mechanism of diguanylate cyclase modulation that might be widespread among bacteria and could control other enzymes in a similar fashion. The present study could form the basis for investigating the function of other DUF1987 domain-containing proteins by providing a first model for an activation mechanism. The work also extends the group's previous finding that SiaC is regulated by phosphorylation, reversing the activation effect.

In summary, the present study is sound and provides new insights into a diguanylate cyclase activation mechanism by a binding partner protein with more general implications for GGDEF regulation and DUF1987 function. The work should be of great interest to a broad audience including the communities interested in cyclic dinucleotide signaling, structural biology, and microbiology.

To further improve the manuscript's quality and clarity, we would recommend addressing the following points. There are three points that, in our opinion, require more effort:

1. The authors conclude that SiaD is a pentamer in the absence of SiaC. This is an important finding since SiaC would need to destabilize this oligomer, stabilizing an alternative, dimeric SiaD conformation, which ultimately poises the GGDEF domains for catalysis. To corroborate this hypothesis, the authors have done an extensive study using SAXS, SEC-MALS, and blue-native PAGE experiments. However, what remains somewhat enigmatic is how SiaC gains access to its binding sites on SiaD, which appear to be non-accessible in the inactive SiaD pentamer. Hence, an improved characterization of SiaD in presence and absence of SiaC is critically important. Specific questions are:

– It is not clear why no SEC-MALS experiments were reported for the SiaC/SiaD complex. Given that the method provided insight into the autoinhibited SiaD oligomer, one may argue that it could also support the formation and activation path (e.g., through SiaC titration experiments) to an active SiaD complex. Given the stoichiometry of the active complex and refolding of the central SiaD stalk, one may also expect cooperativity as part of the SiaC-SiaD mechanism, which was not explored here.

– The authors have performed blue-native PAGE analysis to observe oligomerization of SiaD and SiaD-SiaC complex formation (Figure Supplement 13) and conclude that SiaD may undergo three steps of conformational change upon SiaC binding and activation (i.e., pentamer deoligomerization, reconstruction of the SiaD N-terminal stalk domain, and finally SiaD dimerization with SiaC binding). However, their experiments do not sufficiently substantiate this claim. There are some caveats with this experiment that deserve attention: i. The blue-native PAGE gel lane corresponding to SiaC alone does not show any specific protein? Why is that? What was the concentration of SiaC alone used here? ii. The lane that corresponds to SiaC/SiaD complex does not indicate intermediate species (e.g., SiaD monomer, dimer, trimer or tetramer; or excess SiaC). Thus, the data does not directly support a multi-step process. Titration experiments may provide some resolution of the individual steps of SiaD activation. iii. Some technical details should be reported: How long was the SiaC-SiaD mixture incubated before electrophoresis? What are the kinetics of SiaC-SiaD complex formation? A time-course experiment could provide further insight into the steps leading to SiaD activation and thus providing justification for the proposed mechanism.

2. SEC-SAXS experiments were performed to gain insight into the autoinhibited oligomer of SiaD. Interestingly, the low-resolution SiaD particle model (Figure 4F) would not accommodate a pentameric SiaD with extended stalks as seen in the active-state crystal structure. To fit SiaD into the particle model, the authors have proposed an alternative model of SiaD, which includes a compact conformation of the N-terminal motifs, with the extended stalk domain reduced to five short helices and two β- strands (Figure 4F). Key parameters of how this model was derived were not specified:

– Please share (e.g., in the Material and Methods section) how monomeric SiaD homology modeling was performed (including the modeling of the specific N-terminal fold).

– The authors propose that SiaC induces deoligomerization of the inactive SiaD pentamer and binds to the two conserved motifs in the reconstructed, active SiaD dimer stalk to stabilize its conformation for further catalysis. There are several caveats pertaining to these conclusions. i. No experimental data is provided that demonstrate the loss of two β-strands during SiaC interaction and the increase of helical content. ii. It is unclear how the binding of SiaC helps to fold an extended stalk. One way to potentially clarify this mechanism is to perform Circular Dichroism (CD) to determine the secondary structure and folding properties of SiaD in the presence and absence of SiaC. Mutation of single SiaC-binding sites on SiaD could inform whether there is a particular order of binding, leading to distinct steps in the activation mechanism. For example, it could answer whether there is one particular site on the SiaD stalk that is exposed in the inactive pentamer, where SiaC binding could lead to pentamer disassembly. Such experiments would strengthen the interpretation of the results and provide support for the proposed regulatory mechanism. In general, a more detailed discussion of the activation model is warranted.

3. The fact that diguanylate cyclase activity of SiaD could be inhibited by c-di-GMP (Figure 2—figure supplement 3) suggests the I-sites of SiaD exists and is functional. However, the model has not been confirmed experimentally (e.g., through site-directed mutagenesis) and the structural basis of c-di-GMP binding is not clear as SiaD would have to adopt an alternative conformation for the completion of a proper I-site pocket.

– Specifically, we wonder what a c-di-GMP-inhibited SiaD conformation may look like. Does c-di-GMP revert SiaD back to a pentamer? What is the relative location of the I-sites in the modeled pentamer?

– Some diguanylate cyclases have been reported to be inhibited by product binding to the active site. A targeted mutagenesis study impacting the I-site would confirm that c-di-GMP inhibition is indeed involving the I-site of SiaD. Relevant to this, figure supplement 3 states usage of 50 mM c-di-GMP as inhibitor in the kinetic assays. This seems to be an extremely high concentration. At that concentration, one cannot rule out that the active site is impacted as well (i.e., competition with substrate GTP).

– Is c-di-GMP binding compatible with SiaC binding, or does it destabilize the active state? Answers to these questions could come from assays already established here, for example SEC-MALS and/or SEC-SAXS in the presence of c-di-GMP. Also, the complexes pulled-down by GST could be challenged with c-di-GMP to assess, where the cyclic dinucleotide affects complex stability. Such experiments could be viewed as important additions to the present study as I-site-mediated inhibition is an integral regulatory mechanism in diguanylate cyclases. The analysis seems feasible and could provide more insight into the oligomerization transitions of SiaD.

4. The chapter >The DGC domain of SiaD is critical for substrate binding< (lines 167-229) can be shortened considerably since many high-resolution crystal structures of GGDEF domain-containing proteins have been described in detail in the literature and SiaD follows suit. Likewise, it has already been established that divalent cations bind to a specific site on the GGDEF domains and are important for catalysis. Hence, this entire chapter could be reduced to a few sentences pointing out that SiaD is conserved with regard to these features.

*Reviewer #2:*

The authors detail the mechanism of activation of the diguanylate cyclase SiaD by a partner protein SiaC. This work pertains to the production of cyclic-di-GMP that governs biofilm formation in *Pseudomonas aeruginosa*.

SiaD and related divergent cyclases require a dimeric state to become active, often achieved through stimulus binding to a sensory domain. In this particular case the authors use structural biology and related biochemical experiments to reveal that SiaD activation is driven by SiaC (in its unphosphorylated form) binding to two regions on a helical stalk domain of SiaD. This binding (in a 4:2 stoichiometry) encourages productive dimerization of SiaD (complexed here with a substrate analogue in one of the active sites – an interesting finding but unrelated to the activation process). This model is corroborated by mutants of SiaC (at the SiaC:SiaD interface) unable to bind SiaD and thus unable to form biofilm.

The model additionally benefits from characterisation of non-activated SiaD, via a lower resolution SAXs model, that reveals a pentameric non-catalytic conformation. This does support the above but is more open to interpretation, whereas a higher resolution (atomic) model would make the details of the unactivated state more certain.

The structural biology elements of the manuscript are technically sound and explain the mechanism of activation to a strong degree (hence the authors achieve many of the goals set out to understand this system although a few points remain). The outcome of SiaD stimulation by SiaC represents the first case in the literature of characterization of a cyclase activated by a partner protein; this makes the manuscript of fair interest to the c-di-GMP field, even in the cases where bacteria use a different means of activation.

The SiaC:SiaD model should guide the *Pseudomonas* field and find use in the design of variants to probe c-di-GMP signalling.

I would like the writing / phrasing to be tightened throughout – there are quite a few errors too numerous to list here.

The science therein details activation of a system particular to *Pseudomonas* and related strains – because it is not immediately relevant to a broader audience I would suggest that it would benefit the study to have a higher degree of characterisation. To my mind this could come from one of two points:

– a high resolution form of the unactivated pentameric state that would reveal key contacts (that would be more useful to others in field than a low resolution envelope model).

OR

– mutagenesis of the two different SiaC sites present on SiaD. At present, the mutagenesis focusses on SiaC but instead I was left wondering what the consequences would be if the two SiaC sites on the stalk was reduced to a single site via mutation.

*Reviewer #3:*

This paper seeks to elucidate the structure and mechanism by which the diguanylate cyclase SiaD from the bacterial pathogen *Pseudomonas aeruginosa* is activated by the protein partner switch SiaC. Previously, the same authors showed that SiaC directly activates SiaD to produce cyclic di-GMP, but phosphorylation of SiaC by SiaB represses this activity. The specific questions the authors tackled in this study are: (1) how does SiaC bind to SiaD to activate diguanylate cyclase activity? (2) how does phosphorylation of SiaC prevent binding and activation? (3) what is the inactive conformation of SiaD?

Major strengths include the structure of the SiaD-SiaC complex, which revealed that SiaC binds as a dimer in two places on an α helical 'stalk' of SiaD, which is in an alternate conformation in the absence of SiaC. This may be the first structure of an active diguanylate cyclase enzyme in complex with its protein binding partner. The structure specifically shows how phosphorylation of SiaC prevents binding to SiaD and led to identification of two homologous binding sites for SiaC on the stalk.

Another major strength was the characterization of SiaD. A critical point was determination of the correct coding sequence with additional 40 aa at the N-terminus. SAXS analysis revealed that SiaD forms an inactive pentamer in the absence of SiaC. Throughout, the structural analyses were supported by biochemical methods as well as in vivo biofilm assays of mutants.

One major weakness in the paper is the section on the DGC domain of SiaD. The analysis is written as if there were no prior analyses of GGDEF domains, so is missing a lot of important context that the authors need to include. Another major weakness is there are inconsistencies or gaps in some of the description of the structural analyses of SiaD and SiaC on their own.

The significance of this work relates to the importance of DGC enzymes in regulating cyclic di-GMP levels, which controls biofilm and colonization phenotypes in almost all bacteria. The structure of this complex may serve as a paradigm for other protein partner switch-DGC systems.

1) DGC analysis: It should be made clear whether the observed interactions are the same or different than other GTP bound or GpCpp bound GGDEF structures. Furthermore, the authors did not compare the GpCpp versus no ligand bound GGDEF domains in their structure, and did not analyze the catalytic residues, even though activation of the enzyme is the key feature. Please see https://elifesciences.org/articles/43959: is the conserved general base oriented for catalysis in the SiaD-SiaC complex?

2) For SiaD, please explain how the alternate conformation of the N-terminal region of SiaD in absence of SiaC was determined or modeled.

3) For SiaC, the authors should determine whether it forms a dimer on its own or dimerizes in the presence of SiaD.

4) The authors are inconsistent in their description of SiaC. On one hand, it has a homologous structure to STAS proteins, on the other hand, it contains a domain of unknown function, DUF1987, and has a unique fold. Perhaps the line of reasoning is that their results show that DUF1987 domains are structurally related to STAS domains; if so, this should be more clearly stated.

5) Along these lines, is the N-terminal domain of SiaD a DUF or other conserved domain class? This would strengthen the significance of the paper: if the SiaD-SiaC interaction domains are both conserved, then this structure could serve as the paradigm.

6) The manuscript main text includes a lot of structural description that are mostly listing AA residues, which are redundant with the figures themselves. Please consider revising to focus on reporting the results (what the data mean) rather than describing the data itself. In addition, some of the non-structural figures are made so small that the graph labels are almost illegible.

---

## [Author Response]

Essential revisions:1. A central feature of the activation model is the stabilization of a SiaD dimer by four molecules of SiaC. A mutagenesis study of the two individual SiaC binding sites on SiaD would likely provide insight into the cooperativity of the activation process and may reveal an order of events or relative importance of the binding sites. As readout, enzyme activity (c-di-GMP production) could be correlated with complex formation (via multi-angle laser light scattering). Please also comment whether SiaC is monomeric or dimeric in solution in the absence of SiaD.

Thanks for the good comments. Accordingly, the key residues (S50, Q54 and S82, Q86) of the two SiaC binding sites on SiaD stalk have been mutated into alanine, respectively. Further SEC and SEC-MALS assays showed that both SiaD^S82A-Q86A^ and SiaD^S50A-Q54A^ were able to form complex with SiaC. However, the stability of SiaC-SiaD^S82A-Q86A^ complex turns to be worse and the complex partially dissociated after overnight incubation, indicating that the SiaC binding site of SiaD that is proximal to GGDEF domain (S82, Q86) is more critical for the maintenance of the SiaC-SiaD complex (*Figure 5C*). Furthermore, in vitro activity assays showed that the enzyme activity of SiaD^S82A-Q86A^, but not SiaD^S50A-Q54A^, was significantly lower than that of wild type SiaD in the presence of SiaC (*Figure 3—figure supplement 1*). Consistently, biofilm formation experiment showed that overexpression of SiaD^S50A-Q54A^ restored the biofilm formation of Δ*siaD* mutant. On the contrary, SiaD^S82A-Q86A^ failed to restore the biofilm formation of Δ*siaD* mutant (*Figure 3D*). This result indicated that mutation of the latter SiaC sites influences SiaD activation in vivo. Taken together, these results indicated that the SiaC binding site of SiaD that is proximal to GGDEF domain is functionally more important than the other one.

On the other hand, we considered that there was no certain order for SiaC binding to these two sites on the stalk, for that the amino acid sequences of the two SiaC binding sites are highly conserved (*Figure 4F*), and the ITC experiments showed a continuous “S” curve for SiaC binding to SiaD, indicating that there was no significant difference in affinity between the two binding sites (*Figure 4G*).

Our SEC-MALS assay showed that SiaC was monomeric in solution in the absence of SiaD (*Figure 5—figure supplement 1B*). In addition, our previous GST-pull down assay also have confirmed that GST-SiaC was unable to interact with SiaC-FLAG (Chen et.al. EMBO, 2020). These results demonstrated that SiaC protein is monomeric in solution.

2. A particular regulatory feature of a subset of diguanylate cyclases, including SiaD, is product inhibition via a distinct c-di-GMP binding site. The authors identified such a site on SiaD and demonstrate that c-di-GMP inhibits SiaD (albeit at fairly high second messenger concentration). Considering the central role of inhibitory-site regulation in diguanylate cyclase regulation and c-di-GMP signaling networks, a more thorough analysis of this site on SiaD would be warranted. Minimally, the impact of inhibitory-site mutations on Pseudomonas biofilm formation, SiaD activation and SiaD-SiaC complex stability should be assessed. A central question is whether c-di-GMP binding to the allosteric, inhibitory site destabilizes the SiaC-SiaD complex.

Sorry about for this mistake. The concentration in the reaction buffer was 50 μM, but not 50 mM. We have revised it.

To explore the product inhibition of SiaD, a series of experiments have been performed. SEC-MALS and enzymatic assays showed that binding of c-di-GMP to non-activated SiaD pentamer precludes further interaction of SiaC to SiaD, and thereby inhibits the activation of SiaD (*Figure 3C and figure 3—figure supplement 1*). Moreover, excess of c-di-GMP indeed inhibited, albeit moderately, the enzymatic activity of SiaD-SiaC complex (*Figure 3—figure supplement 1*), although the presence of c-di-GMP does not destabilize the SiaD-SiaC complex (*Figure 3C and figure 3—figure supplement 1*). In consistent with the previous studies on the family homologs, the primary I site mutation of SiaD (SiaD^R201A^) drastically eliminated the inhibitory effects mentioned above as expected (*Figure 3C, D and figure 3—figure supplement 1*). In addition, the protein thermal shift assay (PTSA) showed that while c-di-GMP promotes the SiaD protein stability via binding to the I-site of the DGC domain, and the inactivated SiaD pentamer exhibits a better stability upon c-di-GMP binding than the complex (*Figure 3—figure supplement 2*). These results suggested that there are two distinct inhibitory modes for activated and non-activated SiaD.

Given SiaD homolog PleD has been shown to have two “domain immobilization” modes for product inhibition (In non-activated PleD, the primary I site cross-linked to the secondary I site of the REC2 domain stabilizes an inactive conformation. In activated PleD, dimeric c-di-GMP cross-links the primary I site with a secondary I site on the neighbouring DGC domain to immobilize the active sites away from each other), and the I sites between SiaD and PleD are highly conserved (*Figure 3A*), we deduced that intercalated c-di-GMP dimer inhibits SiaD activity in a manner analogous to that of PleD, and the rotation of the paired GGDEF domains in SiaD-SiaC complex may be facilitated by the soft linkers that connects the N-terminal stalk and GGDEF domains just like PleD (*Figure 6—figure supplement 4*).

3. Central to the model of SiaD activation is the restructuring of the N-terminal domain of the enzyme, from an apparent globular fold containing β-strands to a long, extended helix. However, data supporting this transition is sparse. One way to potentially clarify this mechanism is to perform Circular Dichroism (CD) to determine the secondary structure and folding properties of SiaD in the presence and absence of SiaC.

Thanks. Based on the circular dish-like shape model of SAXS, the structure was predicted by swiss-model (swissmodel.expasy.org). Accordingly, the secondary structure of SiaD in both of the activated or inactivated state was analyzed by far-UV circular dichroism spectroscopy. To characterize the variations of secondary structure for SiaD, we tracked CD signals by gradually increasing SiaC concentration. The CD data suggested that the numbers of α-helix increased, while the numbers of β-sheet decreased during SiaD activation (*Figure 6—figure supplement 3 and table S5*), which supports the model that inactive SiaD adopts a compact N-terminal conformation.

Essential textual changes were summarized elsewhere.Addressing these points would substantially strengthen the conclusion of the study, providing a more complete model of SiaD regulation and bacterial biofilm formation.Reviewer #1:Signaling via the bacterial second messenger c-di-GMP has emerged as a major pathway to control cellular physiology, foremost in the transition from a free-swimming to a sessile, social lifestyle. The production of c-di-GMP in bacteria is catalyzed by GGDEF domain-containing enzymes that act as diguanylate cyclases. Their activity is regulated by various mechanisms, but for many we lack structural insight of how this activity modulation is achieved. The manuscript by Chen et al. describes an interesting molecular mechanism responsible for the activation of a diguanylate cyclase in *P. aeruginosa*, SiaD, by its binding partner protein, SiaC. Based on structural analyses of a catalytically active SiaC-SiaD complex bound to a GTP analog and an autoinhibited SiaD, the authors deduce a mechanism by which four SiaC proteins stabilize an elongated, dimeric conformation of SiaD, positioning the GGDEF domains within the dimer for catalysis. In contrast, the inactive state appears to be a homo-pentamer, with each SiaD protomer adopting a more globular fold. Binding studies and biofilm assays involving structure-guided mutant variants of the proteins further support the proposed mechanism.The observed protein-protein interaction via a protein of unknown function, the DUF1987 domain, may be a diverse mechanism of diguanylate cyclase modulation that might be widespread among bacteria and could control other enzymes in a similar fashion. The present study could form the basis for investigating the function of other DUF1987 domain-containing proteins by providing a first model for an activation mechanism. The work also extends the group's previous finding that SiaC is regulated by phosphorylation, reversing the activation effect.In summary, the present study is sound and provides new insights into a diguanylate cyclase activation mechanism by a binding partner protein with more general implications for GGDEF regulation and DUF1987 function. The work should be of great interest to a broad audience including the communities interested in cyclic dinucleotide signaling, structural biology, and microbiology.To further improve the manuscript's quality and clarity, we would recommend addressing the following points. While these are mostly minor issues, there are three points that, in our opinion, require more effort:

We sincerely thank the positive evaluation for our manuscript. The constructive comments and suggestions have helped us improve the quantity of our manuscript.

1. The authors conclude that SiaD is a pentamer in the absence of SiaC. This is an important finding since SiaC would need to destabilize this oligomer, stabilizing an alternative, dimeric SiaD conformation, which ultimately poises the GGDEF domains for catalysis. To corroborate this hypothesis, the authors have done an extensive study using SAXS, SEC-MALS, and blue-native PAGE experiments. However, what remains somewhat enigmatic is how SiaC gains access to its binding sites on SiaD, which appear to be non-accessible in the inactive SiaD pentamer. Hence, an improved characterization of SiaD in presence and absence of SiaC is critically important. Specific questions are:– It is not clear why no SEC-MALS experiments were reported for the SiaC/SiaD complex. Given that the method provided insight into the autoinhibited SiaD oligomer, one may argue that it could also support the formation and activation path (e.g., through SiaC titration experiments) to an active SiaD complex. Given the stoichiometry of the active complex and refolding of the central SiaD stalk, one may also expect cooperativity as part of the SiaC-SiaD mechanism, which was not explored here.

Thanks. We had supplemented the SEC-MALS results of SiaC-SiaD complex. The results showed that the molecular weight of the complex was 120 kD, which was completely matched with the data of SAXS and crystal structure (*Figure 5—figure supplement 1A*).

To identify the role of the two SiaC-binding sites during SiaD activation and to get insight into the formation and activation path to an active SiaD complex, the key residues (S50, Q54 and S82, Q86) of the two SiaC binding sites on SiaD stalk have been mutated into alanine, respectively. Further SEC and SEC-MALS assays showed that both SiaD^S82A-Q86A^ and SiaD^S50A-Q54A^ were able to form complex with SiaC. However, the stability of SiaC-SiaD^S82A-Q86A^ complex turns to be worse and the complex partially dissociated after overnight incubation, indicating that the SiaC binding site of SiaD that is proximal to GGDEF domain (S82, Q86) is more critical for the maintenance of the SiaC-SiaD complex (*Figure 5C*). Furthermore, in vitro activity assays showed that the enzyme activity of SiaD^S82A-Q86A^, but not SiaD^S50A-Q54A^, was significantly lower than that of wild type SiaD in the presence of SiaC (*Figure 3—figure supplement 1*). Consistently, biofilm formation experiment showed that overexpression of SiaD^S50A-Q54A^ restored the biofilm formation of Δ*siaD* mutant. On the contrary, SiaD^S82A-Q86A^ failed to restore the biofilm formation of Δ*siaD* mutant (*Figure 3D*). This result indicated that mutation of the latter SiaC sites influences SiaD activation in vivo. Taken together, these results indicated that the SiaC binding site of SiaD that is proximal to GGDEF domain is functionally more important than the other one.

– The authors have performed blue-native PAGE analysis to observe oligomerization of SiaD and SiaD-SiaC complex formation (Figure Supplement 13) and conclude that SiaD may undergo three steps of conformational change upon SiaC binding and activation (i.e., pentamer deoligomerization, reconstruction of the SiaD N-terminal stalk domain, and finally SiaD dimerization with SiaC binding). However, their experiments do not sufficiently substantiate this claim. There are some caveats with this experiment that deserve attention: i. The blue-native PAGE gel lane corresponding to SiaC alone does not show any specific protein? Why is that? What was the concentration of SiaC alone used here?

The concentration of SiaC alone was 35 μg. The result for SiaC after electrophoresis was showed in the revised manuscript (*Figure 6—figure supplement 2*).

ii. The lane that corresponds to SiaC/SiaD complex does not indicate intermediate species (e.g., SiaD monomer, dimer, trimer or tetramer; or excess SiaC). Thus, the data does not directly support a multi-step process. Titration experiments may provide some resolution of the individual steps of SiaD activation.

Thanks for the suggestive comments. Our blue-native result could not directly support a multi-step process during SiaD activation. To get insight into the process for SiaD activation, we performed circular dichroism (*Figure 6—figure supplement 2,* see below please).

iii. Some technical details should be reported: How long was the SiaC-SiaD mixture incubated before electrophoresis? What are the kinetics of SiaC-SiaD complex formation? A time-course experiment could provide further insight into the steps leading to SiaD activation and thus providing justification for the proposed mechanism.

Thanks for these constructive suggestions. The SiaD-SiaC was incubated for 30 minutes before electrophoresis. To get insight into the process for SiaD activation beside from the blue-PAGE data, we performed a series of circular dichroism spectrum experiments by gradually increasing the concentration of SiaC from 0 to 16 μM (Materials and methods). The CD data suggested that the numbers of α-helix increased, while the numbers of β-sheet decreased during SiaD activation (Figure 6-figure 6—figure supplement 3 and table S5), which supports the model that inactivated SiaD adopts a more compact N-terminal conformation. In addition, the ITC titration experiments showed a continuous “S” curve for SiaC-SiaD binding, indicating that there was no significant difference in affinity between the two binding sites (Figure 4G). we considered that there was no certain order for SiaC binding to these two sites on the stalk.

2. SEC-SAXS experiments were performed to gain insight into the autoinhibited oligomer of SiaD. Interestingly, the low-resolution SiaD particle model (Figure 4F) would not accommodate a pentameric SiaD with extended stalks as seen in the active-state crystal structure. To fit SiaD into the particle model, the authors have proposed an alternative model of SiaD, which includes a compact conformation of the N-terminal motifs, with the extended stalk domain reduced to five short helices and two β- strands (Figure 4F). Key parameters of how this model was derived were not specified:– Please share (e.g., in the Material and Methods section) how monomeric SiaD homology modeling was performed (including the modeling of the specific N-terminal fold).

Thanks. Based on the circular dish-like shape model of SAXS, the structure was predicted by swiss-model (swissmodel.expasy.org).

– The authors propose that SiaC induces deoligomerization of the inactive SiaD pentamer and binds to the two conserved motifs in the reconstructed, active SiaD dimer stalk to stabilize its conformation for further catalysis. There are several caveats pertaining to these conclusions. i. No experimental data is provided that demonstrate the loss of two β-strands during SiaC interaction and the increase of helical content. ii. It is unclear how the binding of SiaC helps to fold an extended stalk. One way to potentially clarify this mechanism is to perform Circular Dichroism (CD) to determine the secondary structure and folding properties of SiaD in the presence and absence of SiaC.

Thanks for your good suggestions. Accordingly, the secondary structure of SiaD in its activated or inactivated state was analyzed by far-UV circular dichroism spectroscopy. To characterize the variations of secondary structure for SiaD, we tracked CD signals by gradually increasing SiaC concentration. The CD data suggested that the numbers of α-helix increased, while the numbers of β-sheet decreased during SiaD activation (Figure 6-figure 6—figure supplement 3 and table S5), which supports the model that inactive SiaD adopts a compact N-terminal conformation.

Mutation of single SiaC-binding sites on SiaD could inform whether there is a particular order of binding, leading to distinct steps in the activation mechanism. For example, it could answer whether there is one particular site on the SiaD stalk that is exposed in the inactive pentamer, where SiaC binding could lead to pentamer disassembly. Such experiments would strengthen the interpretation of the results and provide support for the proposed regulatory mechanism. In general, a more detailed discussion of the activation model is warranted.

Thanks. Accordingly, the key residues (S50, Q54 and S82, Q86) of the two SiaC binding sites on SiaD stalk have been mutated into alanine, respectively. Further SEC and SEC-MALS assays showed that both SiaD^S82A-Q86A^ and SiaD^S50A-Q54A^ were able to form complex with SiaC. However, the stability of SiaC-SiaD^S82A-Q86A^ complex turns to be worse and the complex partially dissociated after overnight incubation, indicating that the SiaC binding site of SiaD that is proximal to GGDEF domain (S82, Q86) is more critical for the maintenance of the SiaC-SiaD complex (*Figure 5C*). Furthermore, in vitro activity assays showed that the enzyme activity of SiaD^S82A-Q86A^, but not SiaD^S50A-Q54A^, was significantly lower than that of wild type SiaD in the presence of SiaC (*Figure 3—figure supplement 1*). Consistently, biofilm formation experiment showed that overexpression of SiaD^S50A-Q54A^ restored the biofilm formation of Δ*siaD* mutant. On the contrary, SiaD^S82A-Q86A^ failed to restore the biofilm formation of Δ*siaD* mutant (*Figure 3D*). This result indicated that mutation of the latter SiaC sites influences SiaD activation in vivo. Taken together, these results indicated that the SiaC binding site of SiaD that is proximal to GGDEF domain is functionally more important than the other one.

3. The fact that diguanylate cyclase activity of SiaD could be inhibited by c-di-GMP (Figure 2—figure supplement 3) suggests the I-sites of SiaD exists and is functional. However, the model has not been confirmed experimentally (e.g., through site-directed mutagenesis) and the structural basis of c-di-GMP binding is not clear as SiaD would have to adopt an alternative conformation for the completion of a proper I-site pocket.– Specifically, we wonder what a c-di-GMP-inhibited SiaD conformation may look like. Does c-di-GMP revert SiaD back to a pentamer? What is the relative location of the I-sites in the modeled pentamer?

Thanks. To explore the product inhibition of SiaD, a series of experiments have been performed. SEC-MALS and enzymatic assays showed that binding of c-di-GMP to non-activated SiaD pentamer precludes further interaction of SiaC to SiaD, and thereby inhibits the activation of SiaD (*Figure 3C and figure 3—figure supplement 1*). Moreover, excess of c-di-GMP indeed inhibited, albeit moderately, the enzymatic activity of SiaD-SiaC complex (*Figure 3—figure supplement 1*), although the presence of c-di-GMP does not destabilize the SiaD-SiaC complex (*Figure 3C and figure 3—figure supplement 1*). In consistent with the previous studies on the family homologs, the primary I site mutation of SiaD (SiaD^R201A^) drastically eliminated the inhibitory effects mentioned above as expected (*Figure 3C, D and figure 3—figure supplement 1*). In addition, the protein thermal shift assay (PTSA) showed that while c-di-GMP promotes the SiaD protein stability via binding to the I-site of the DGC domain, and the inactivated SiaD pentamer exhibits a better stability upon c-di-GMP binding than the complex (*Figure 3—figure supplement 2*). These results suggested that there are two distinct inhibitory modes for activated and non-activated SiaD.

Given SiaD homolog PleD has been shown to have two “domain immobilization” modes for product inhibition (In non-activated PleD, the primary I site cross-linked to the secondary I site of the REC2 domain stabilizes an inactive conformation. In activated PleD, dimeric c-di-GMP cross-links the primary I site with a secondary I site on the neighbouring DGC domain to immobilize the active sites away from each other), and the I sites between SiaD and PleD are highly conserved (*Figure 3A*), we deduced that intercalated c-di-GMP dimer inhibits SiaD activity in a manner analogous to that of PleD, and the rotation of the paired GGDEF domains in SiaD-SiaC complex may be facilitated by the soft linkers that connects the N-terminal stalk and GGDEF domains just like PleD (*Figure 6—figure supplement 4*).

– Some diguanylate cyclases have been reported to be inhibited by product binding to the active site. A targeted mutagenesis study impacting the I-site would confirm that c-di-GMP inhibition is indeed involving the I-site of SiaD. Relevant to this, figure supplement 3 states usage of 50 mM c-di-GMP as inhibitor in the kinetic assays. This seems to be an extremely high concentration. At that concentration, one cannot rule out that the active site is impacted as well (i.e., competition with substrate GTP).

Thanks. Sorry about for this mistake. The concentration in the reaction buffer was 50 μM, but not 50 mM. We have corrected it.

Accordingly, we have constructed mutations of primary I site residues to explore the role of I-site during SiaD regulation. Although mutation of residue R170 was deleterious for activity, we previously revealed that mutation R170G significantly promoted biofilm formation (*Chen et.al.,* EMBO, 2020, 39(6):e103412). We revealed that I-site mutation (R201A) significantly promoted biofilm formation (*Figure 3D*). Consistently, R201A mutation obviously promoted SiaD activity even in the presence of excess c-di-GMP (*Figure 6—figure supplement 4*). SEC results showed that binding of c-di-GMP to SiaD abolishes formation of a 2:4 stoichiometric SiaD:SiaC complex, which is dependent on the residue R201 located on I-site. These results demonstrated that the I-site plays a central role during c-di-GMP-mediated inhibition.

– Is c-di-GMP binding compatible with SiaC binding, or does it destabilize the active state? Answers to these questions could come from assays already established here, for example SEC-MALS and/or SEC-SAXS in the presence of c-di-GMP. Also, the complexes pulled-down by GST could be challenged with c-di-GMP to assess, where the cyclic dinucleotide affects complex stability. Such experiments could be viewed as important additions to the present study as I-site-mediated inhibition is an integral regulatory mechanism in diguanylate cyclases. The analysis seems feasible and could provide more insight into the oligomerization transitions of SiaD.

Thanks for your comments. SEC-MALS and enzymatic assays showed that binding of c-di-GMP to non-activated SiaD pentamer precludes further interaction of SiaC to SiaD, and thereby inhibits the activation of SiaD (*Figure 3C and figure 3—figure supplement 1*). Moreover,excess of c-di-GMP indeed inhibited, albeit moderately, the enzymatic activity of SiaD-SiaC complex (*Figure 3—figure supplement 1*), although the presence of c-di-GMP does not destabilize the SiaD-SiaC complex (*Figure 3C and figure 3—figure supplement 1*). In consistent with the previous studies on the family homologs, the primary I site mutation of SiaD (SiaD^R201A^) drastically eliminated the inhibitory effects mentioned above as expected (*Figure 3C, D and figure 3—figure supplement 1*).

4. The chapter >The DGC domain of SiaD is critical for substrate binding< (lines 167-229) can be shortened considerably since many high-resolution crystal structures of GGDEF domain-containing proteins have been described in detail in the literature and SiaD follows suit. Likewise, it has already been established that divalent cations bind to a specific site on the GGDEF domains and are important for catalysis. Hence, this entire chapter could be reduced to a few sentences pointing out that SiaD is conserved with regard to these features.

Thanks. We have condensed this chapter into a few sentences which point out that SiaD is conserved with regard to these features. Furthermore, the content regarding the product inhibition of SiaD was expanded.

Reviewer #2:The authors detail the mechanism of activation of the diguanylate cyclase SiaD by a partner protein SiaC. This work pertains to the production of cyclic-di-GMP that governs biofilm formation in *Pseudomonas aeruginosa*.SiaD and related divergent cyclases require a dimeric state to become active, often achieved through stimulus binding to a sensory domain. In this particular case the authors use structural biology and related biochemical experiments to reveal that SiaD activation is driven by SiaC (in its unphosphorylated form) binding to two regions on a helical stalk domain of SiaD. This binding (in a 4:2 stoichiometry) encourages productive dimerization of SiaD (complexed here with a substrate analogue in one of the active sites – an interesting finding but unrelated to the activation process). This model is corroborated by mutants of SiaC (at the SiaC:SiaD interface) unable to bind SiaD and thus unable to form biofilm.The model additionally benefits from characterisation of non-activated SiaD, via a lower resolution SAXs model, that reveals a pentameric non-catalytic conformation. This does support the above but is more open to interpretation, whereas a higher resolution (atomic) model would make the details of the unactivated state more certain.The structural biology elements of the manuscript are technically sound and explain the mechanism of activation to a strong degree (hence the authors achieve many of the goals set out to understand this system although a few points remain). The outcome of SiaD stimulation by SiaC represents the first case in the literature of characterization of a cyclase activated by a partner protein; this makes the manuscript of fair interest to the c-di-GMP field, even in the cases where bacteria use a different means of activation.The SiaC:SiaD model should guide the Pseudomonas field and find use in the design of variants to probe c-di-GMP signalling.

We sincerely thank the positive evaluation for our manuscript. The constructive comments and suggestions have helped us improve the quantity of our manuscript.

I would like the writing / phrasing to be tightened throughout – there are quite a few errors too numerous to list here.

Thanks. We have carefully corrected these errors and rearranged some sentences in the revised manuscript.

The science therein details activation of a system particular to Pseudomonas and related strains – because it is not immediately relevant to a broader audience I would suggest that it would benefit the study to have a higher degree of characterisation. To my mind this could come from one of two points:– a high resolution form of the unactivated pentameric state that would reveal key contacts (that would be more useful to others in field than a low resolution envelope model).OR– mutagenesis of the two different SiaC sites present on SiaD. At present, the mutagenesis focusses on SiaC but instead I was left wondering what the consequences would be if the two SiaC sites on the stalk was reduced to a single site via mutation.

Thanks for the good suggestions. Accordingly, the key residues (S50, Q54 and S82, Q86) of the two SiaC binding sites on SiaD stalk have been mutated into alanine, respectively. Further SEC and SEC-MALS assays showed that both SiaD^S82A-Q86A^ and SiaD^S50A-Q54A^ were able to form complex with SiaC. However, the stability of SiaC-SiaD^S82A-Q86A^ complex turns to be worse and the complex partially dissociated after overnight incubation, indicating that the SiaC binding site of SiaD that is proximal to GGDEF domain (S82, Q86) is more critical for the maintenance of the SiaC-SiaD complex (*Figure 5C*). Furthermore, in vitro activity assays showed that the enzyme activity of SiaD^S82A-Q86A^, but not SiaD^S50A-Q54A^, was significantly lower than that of wild type SiaD in the presence of SiaC (*Figure 3—figure supplement 1*). Consistently, biofilm formation experiment showed that overexpression of SiaD^S50A-Q54A^ restored the biofilm formation of Δ*siaD* mutant. On the contrary, SiaD^S82A-Q86A^ failed to restore the biofilm formation of Δ*siaD* mutant (*Figure 3D*). This result indicated that mutation of the latter SiaC sites influences SiaD activation in vivo. Taken together, these results indicated that the SiaC binding site of SiaD that is proximal to GGDEF domain is functionally more important than the other one.

Reviewer #3:This paper seeks to elucidate the structure and mechanism by which the diguanylate cyclase SiaD from the bacterial pathogen *Pseudomonas aeruginosa* is activated by the protein partner switch SiaC. Previously, the same authors showed that SiaC directly activates SiaD to produce cyclic di-GMP, but phosphorylation of SiaC by SiaB represses this activity. The specific questions the authors tackled in this study are: (1) how does SiaC bind to SiaD to activate diguanylate cyclase activity? (2) how does phosphorylation of SiaC prevent binding and activation? (3) what is the inactive conformation of SiaD?Major strengths include the structure of the SiaD-SiaC complex, which revealed that SiaC binds as a dimer in two places on an α helical 'stalk' of SiaD, which is in an alternate conformation in the absence of SiaC. This may be the first structure of an active diguanylate cyclase enzyme in complex with its protein binding partner. The structure specifically shows how phosphorylation of SiaC prevents binding to SiaD and led to identification of two homologous binding sites for SiaC on the stalk.

We sincerely thank the positive evaluation for our manuscript. The constructive comments and suggestions have helped us improve the quantity of our manuscript.

Another major strength was the characterization of SiaD. A critical point was determination of the correct coding sequence with additional 40 aa at the N-terminus. SAXS analysis revealed that SiaD forms an inactive pentamer in the absence of SiaC. Throughout, the structural analyses were supported by biochemical methods as well as in vivo biofilm assays of mutants.One major weakness in the paper is the section on the DGC domain of SiaD. The analysis is written as if there were no prior analyses of GGDEF domains, so is missing a lot of important context that the authors need to include.

Thanks. We have revised this section. The statements about the canonical features of GGDEF have been condensed into several sentences. And contents about I-site regulatory role were added.

Another major weakness is there are inconsistencies or gaps in some of the description of the structural analyses of SiaD and SiaC on their own.

Thanks. We had unified the description of SiaD and SiaC (P17, Lines: 462-463, 465, 476).

The significance of this work relates to the importance of DGC enzymes in regulating cyclic di-GMP levels, which controls biofilm and colonization phenotypes in almost all bacteria. The structure of this complex may serve as a paradigm for other protein partner switch-DGC systems.1) DGC analysis: It should be made clear whether the observed interactions are the same or different than other GTP bound or GpCpp bound GGDEF structures.

Thanks. The interactions are the same with other GTP bound GGD(E)EF structures such as PleD (*Figure 2—figure supplement 3A*). DGC domain is highly conservative and has been clearly studied, so we have reduced the description of this paragraph.

Furthermore, the authors did not compare the GpCpp versus no ligand bound GGDEF domains in their structure, and did not analyze the catalytic residues, even though activation of the enzyme is the key feature.

Thanks for your comments. Comparing the structure of SiaD-A (with GpCpp) and SiaD-B, there are conformational deflections of several catalytic amino acids (D138, E181, K250 and R254), which further stabilize the binding of Mg^2+^ and the phosphate group of GpCpp(*Figure 2—figure supplement 3*). The role of several catalytic residues, such as D138, D155, E181, K250 and R254, has been evaluated by biofilm formation assay and enzymatic assay. Data showed that these residues are essential for SiaD activity.

Please see https://elifesciences.org/articles/43959: is the conserved general base oriented for catalysis in the SiaD-SiaC complex?

By superposition with *Is*PadC (pdb code: 5LLX), we observed that the conserved general base E182 (E375 in *Gm*GacA) is oriented for catalysis in the SiaC-SiaD complex, which suggested the cooperativity and conservation of SiaD as a catalytic dimer(*Figure 2—figure supplement 3B*).

2) For SiaD, please explain how the alternate conformation of the N-terminal region of SiaD in absence of SiaC was determined or modeled.

Thanks. Based on the circular dish-like shape model of SAXS, the structure was predicted by swiss-model (swissmodel.expasy.org).

3) For SiaC, the authors should determine whether it forms a dimer on its own or dimerizes in the presence of SiaD.

Our SEC-MALS and GST-pull data showed that SiaC was monomeric on its own. In the complex, SiaC binds to the conserved sites of SiaD, and the distance between SiaC is not enough to form dimer.

4) The authors are inconsistent in their description of SiaC. On one hand, it has a homologous structure to STAS proteins, on the other hand, it contains a domain of unknown function, DUF1987, and has a unique fold. Perhaps the line of reasoning is that their results show that DUF1987 domains are structurally related to STAS domains; if so, this should be more clearly stated.

Thanks. We have clearly stated the relevant description in the revised manuscript (P17, Lines: 477).

5) Along these lines, is the N-terminal domain of SiaD a DUF or other conserved domain class? This would strengthen the significance of the paper: if the SiaD-SiaC interaction domains are both conserved, then this structure could serve as the paradigm.

Thanks. The DUF1987 domains of SiaC were widely distributed. However, analysis of the SiaD N-terminal sequence by BLAST and InterPro revealed no conserved domain class.

6) The manuscript main text includes a lot of structural description that are mostly listing AA residues, which are redundant with the figures themselves. Please consider revising to focus on reporting the results (what the data mean) rather than describing the data itself.

Many structural description sentences and paragraphs have been rearranged accordingly.

In addition, some of the non-structural figures are made so small that the graph labels are almost illegible.

Thanks. We have rearranged these figures.